# A Comparison of Drought Information in Early North American Colonial Documentary Records and a High-Resolution Tree Ring-Based Reconstruction

Sam White[1]

[1]Department of History, Ohio State University, Columbus, OH, 43125, USA

*Correspondence to*: Sam White (white.2426@osu.edu)

**Abstract.** Historical documentary records contain valuable information on climate, weather, and their societal impacts during the pre-instrumental period, but it may be difficult to assess the objectivity and reliability of this information, particularly where the documentary record is incomplete or the reliability of the information it contains is uncertain. This article presents a comprehensive review of information relating to drought found in original written records concerning all early European expeditions (1510-1610CE) into the present US and Canada, and compares this information with maps and time series of drought generated from the tree ring-based North American Drought Atlas (NADA). The two sources mostly agree in the timing and location of droughts. This correspondence suggests that much of the information in these early colonial historical records is probably objective and reliable and that tree ring-based drought atlases can provide information relevant to local and regional human historical events, at least in locations where their reconstruction skill is particularly high. This review of drought information from written sources and tree ring-based reconstructions also highlights the extraordinary challenges faced by early European explorers and colonists in North America due to climatic variability in an already unfamiliar and challenging environment.

## 1 Introduction

Regional climatic variability before the instrumental period, including the frequency and severity of historical droughts, may be reconstructed through analysis of proxies preserved in natural archives (paleoclimatology) or of proxies and descriptions in written records (historical climatology). The latter approach has been employed with high reconstruction skill for regions and periods with abundant and consistent personal writings and official archives, including parts of Europe and China during the past six centuries. However, most historical regions and periods lack such abundant and consistent written records before the instrumental period. In these cases, historical observers tend to act as a "high-pass filter" recording mostly extreme events and/or as degraded archives preserving scattered periodic climate proxies and descriptions (Bradley, 2015:519-520). Nevertheless, these written records remain potentially valuable for climate history since they contain information unavailable from proxies in natural archives, including precisely located and dated climate and weather events from all seasons and their societal impacts (Brönniman et al., 2018).

The challenge of using information in historical written records becomes particularly acute in the case of early European overseas exploration and colonization, including overseas imperial exploration and invasion, early religious missions, merchant companies, and the slave trade. On the one hand, these activities left written records of major importance to climate and environmental history. For North America, Australia, and much of Africa and Latin America, they provide the earliest written records of any kind, including observations of climate and weather conditions as well as climatic impacts on indigenous societies and their adaptations. Early colonial observers were typically much more sensitive to climatic and environmental conditions than observers in their home countries and therefore more likely to record and disseminate information on these topics. On the other hand, these same observers were unfamiliar with their new colonial environments and climates and at first lacked institutions and activities that could systematically record climate proxies and descriptions. The resulting records are typically too inconsistent to produce independent verifiable reconstructions of climate parameters. Even in cases where colonial activities continued into later periods and began to leave consistent records appropriate for

standard historical climatology analysis and reconstruction—e.g., indices—one cannot assume that observations made during the unique conditions of these first colonial experiences were equally reliable or objective as those made in later years and could therefore be calibrated in the same manner for climate reconstruction purposes (Pfister et al., 2018; Dobrovolný, 2018).

An alternative approach to examining the accuracy of climatic and other environmental information in these early
colonial records is to compare their information on a defined parameter with a high-resolution reconstruction of that parameter based on proxies in a natural archive. This approach is particular promising in the case of early colonial North America, here defined as the present United States and Canada 1510-1610CE. Colonial activities during this period left a large, diverse, well-preserved, and accessible body of documentary evidence with considerable information on environmental and climatic conditions (section 2.1). This information may be compared with the North American Drought Atlas: a highly spatially and
temporally resolved gridded Palmer Drought Severity Index (PDSI) reconstruction covering the region and period under study (section 2.2). Moreover, the NADA reconstruction indicates sharply contrasting wet and dry conditions during different colonial expeditions in this period, which should highlight (in)consistencies between information in the documentary record and the tree ring-based reconstruction. As previous scholars have concluded, "the integration of the [NADA] PDSI reconstructions with historical information on environmental conditions and the activities of Euroamerican and Native
American societies is a largely unexploited opportunity that promises significant new insight into American history and environmental change" (Stahle et al., 2007).

Thus, this study makes a systematic comparison of information concerning drought from the documentary record of early colonial America with the NADA PDSI reconstruction. The primary goal of this comparison is to examine the objectivity and reliability of climate and environmental information in early European colonial records, and thus their potential use for
historical climatology and environmental history. In addition, this study may assist the following goals: (1) to crosscheck the NADA reconstructions—including those for extreme events reconstructed during the 16th and early 17th centuries—and the NADA's applicability to the scale of local human historical events; (2) to gain further insights into the seasonality and severity of droughts in early colonial North America by combining the information from natural and human archives; and (3) to better understand the human impacts of climate variability during this critical and vulnerable phase of North American exploration
and colonization.

This study acknowledges three important limitations. First, the NADA is primarily a reconstruction of soil moisture deficit (hydrological drought) while human observers mainly recorded deficits of rainfall (meteorological drought) and failures of crops attributed to lack of water (agricultural drought). Second, most written observations about climate and weather in early North American colonial records concern temperature and storms rather than drought. While it may be reasonable to
infer that observations about drought were no more or less reliable and objective than observations about other climate and environmental conditions, there is no similarly robust and highly resolved proxy-based reconstruction of temperature or other parameters with which to make a similar comparison. Third, the NADA cannot be regarded as a perfect reconstruction of drought conditions, and therefore discrepancy between NADA and the documentary record does not necessarily mean that one is "right" and the other "wrong." Nevertheless, frequent disagreement between the documentary record and the NADA
reconstruction should raise concerns about the reliability of the former or the precision of the latter, while consistent agreement should build confidence in both.

A recent review has identified North America as one of the least studied regions for the historical climatology of drought (Brázdil et al., 2018). Although there are abundant personal and official records dating back to the colonial period, many of the them containing information on weather and climate, few researchers have used them to reconstruct the frequency
or severity of historical droughts (Mock, 2012; White, 2018). Archaeologists have reviewed physical and written evidence for the impact of climate and extreme weather, including droughts, during the first century of European expeditions in several regional contexts, particularly the Chesapeake (Blanton, 2000; Blanton, 2003; Blanton, 2004; Rockman, 2010), the southeastern US (Burnett and Murray, 1993), Florida (Paar, 2009; Blanton, 2013), and New Mexico (van West et al., 2013).

A few historians have begun to incorporate climatic perspectives into accounts of early North American exploration and colonies (Kupperman, 2007a; Grandjean, 2011; Wickman, 2015; Wickman, 2018), and a 2017 monograph has provided a comprehensive narrative of the role of regional climate differences and climatic variability in early Spanish, French, and English exploration and colonization of the present US and Canada (White, 2017). These studies indicate that climatic factors,

including drought, played a major role in the failure of several early colonial expeditions in North America and the discouragement or redirection of others, as well as conflict between European colonies and Native Americans. This article provides the first concise comprehensive overview of the evidence concerning drought from historical records and systematic comparison with the NADA drought reconstructions.

**2. Sources and Methods**

**2.1. The Documentary Record**

A search was made for all primary historical records surviving from or directly relevant to every documented European colonial expedition that spent time in the present US and Canada from 1510-1610 CE. This search resulted in several thousand documents contained in several hundred printed volumes, as well as electronic collections and unpublished archival series in

the Archivo General de Indias (Spain) and National Archives (UK), mostly available online (see Supplementary Material). The author read each of these personally and, where possible, in their entirety. Some documents were consulted in English translations, but in those cases almost all descriptions of weather, climate, or climatic impacts were read again in their original languages (French, Spanish, German, Italian, Latin, and Dutch) to correct any mistranslations and to ensure accuracy. All dates were converted from the Julian calendar into modern calendar dates with the new year beginning on January 1.

Descriptions of weather, climate, and climatic impacts were not simply extracted from their context and compiled separately but always considered contextually, meaning that the study took into account: (1) the description's context within the individual document and its larger corpus; (2) the conditions of the observation and of its recording and publication; (3) the background of the author or authors, including their subjective sense of 'normal' weather or climate; and (4) the genre of writing in which any description appeared. (For a complete book-length account of the historical context of climate

observations in the course of early exploration and colonies, see White, 2017.)

The exploration and colonization of North America proceeded in sporadic, uncoordinated ventures by Spanish, French, and English companies until 1610CE, by which date enduring colonies were established at St. Augustine (Florida), Santa Fe (New Mexico), Jamestown (Virginia), and Quebec (Quebec). Written first-hand observations potentially capable of providing useful information concerning the presence, absence, or impacts of droughts were left by members of colonial

expeditions for the following regions and years: the southeastern US and Gulf Coast, principally Florida (1526, 1528-29, 1539-43, 1559-60, 1560, 1562-1610); the southwestern US, principally New Mexico (1530-35, 1540-41, 1581, 1583, 1590, 1598-1610); Virginia and North Carolina (1570, 1585-88, 1607-1610); the California coast (1542, 1587, 1595, 1596, 1602-03); the New England coast (1524, 1602-05, 1607-08); Nova Scotia (1606-07); and Quebec (1600-01, 1608-10).

The relevant surviving records fall broadly into six genres: private correspondence, official correspondence and

memoranda, correspondence of religious orders, pamphlets, travelogues, and finally chronicles and other second-hand compilations of information, which were used only to supplement first-hand information. Each of these genres has been used in historical climatology and each presents particular strengths and weaknesses as a source of weather, climate, and impact evidence (Pfister, 2018; Pfister and White, 2018). However, the unique conditions under which early colonial records were compiled endow them with peculiar advantages and disadvantages compared to other similar written records. On the one

hand, events during the exploration of new lands and the colonization of new territories were unusually well recorded for the period, and their records have been carefully sought out, preserved, and analyzed by scholars concerned with the historical and environmental significance of colonization. Moreover, observers present in unfamiliar environments and concerned with prospects for resource extraction, trade, colonization, or missionization usually devoted more attention to environmental

features, including weather and climate, than those at home (Taylor, 1993; White, 2015b; Zilberstein, 2016). Europeans on early colonial expeditions were often acutely vulnerable to climatic variability and extremes. On the other hand, the earliest colonial observations in each location necessarily predate the establishment of local institutions or practices that could produce regular records containing climate proxies, such as officially prescribed annual harvest dates. Furthermore, the novelty of

colonial environments made it difficult for observers to determine what was normal weather or climate or to identify reliable phenological markers of variability or extremes. Adding to their confusion, early European explorers and colonists of North America expected climates to align with latitudes, overlooking the differences between Europe's predominately maritime climates and North America's predominately continental ones (Kupperman, 1982; Rockman, 2010). This study aims to evaluate whether the advantages of these sources outweigh their potential drawbacks when it comes to reconstructing past

weather and climate.

This study identified four main types of evidence concerning drought for early colonial North America: (1) *Phenological descriptions*, including the growth of plants and conditions of rivers, provide the most objective indicators of drought, but they are not common in all sources. (2) *Narrative descriptions*, describing a lack of precipitation are found more frequently but provide less certain indicators of the presence or absence of drought. (3) *Societal impacts*, particularly crop

failures or famine attributed to drought, appear most frequently in the sources, and may provide confirmation that conditions were unusually severe. However, these descriptions need to be considered in the context of societal vulnerabilities, which varied according to the colony or Native American society in question. Most of the populations discussed in this study relied heavily on crops of maize, which in turn depended on adequate summer rainfall and a lengthy frost-free growing season. (4) *Rain-making ceremonies* represent a peculiar but potentially valuable type of evidence for drought. Measuring the occurrence

and scale of officially sanctioned rain prayers, known as rogation ceremonies, has been demonstrated as a reliable method of drought reconstruction in Spain and Spanish America (Domínguez Castro et al., 2008; Domínguez Castro et al., 2018). Historians have identified similar ceremonies outside Spain in many different religious and cultural traditions, but these have not yet been tested in the same fashion. In the colonial North American context, many early European expeditions reported performing rain prayers, being asked by Native Americans to perform rain prayers, or observing indigenous communities

performing rain-making ceremonies. Some historians have argued that such accounts could have been biased or even falsified by early European observers eager to present Native Americans as simple pagans ready to be won over to Christianity. However, a close review of the context and consistency of these accounts across sources has indicated that most were probably based on actual events (Kupperman, 2007b; White, 2015a).

Information concerning drought was sorted by each type of evidence. It was evaluated according to whether the

information is based on a single observer, two or three observers, or many observers in agreement, and according to whether it provides a definite or indefinite impression of drought. For instance, an observation that it had not rained for a certain number of months was taken as a definite impression, whereas a general description of dry weather was taken as indefinite. Results and further information for specific expeditions are provided in section 3.2.

**2.2. Reconstructions from Proxies in Natural Archives**

Most high-resolution reconstructions of drought in North America for the pre-instrumental period have relied on measures of the variation in the width of growth rings in trees whose growth has been limited by soil moisture during the growing season. The NADA is a set of annual June-August PDSI reconstructions based on 1,845 tree-ring chronologies estimated on a 0.5° latitude/longitude grid, which covers most of North America for the past several centuries (Cook et al.,

1999; Cook et al., 2010). The PDSI reconstructed in NADA is a function of precipitation and evapotranspiration from the previous winter through the summer growing season. The NADA has been found to have a dominant winter precipitation signal in the Southwest and Florida and a dominant summer signal in most of the rest of the continent; however, the signal at any given location may depend on local conditions and the proximity of tree-ring samples (St. George et al., 2010). For this

reason, variations in PDSI over large regions reconstructed by the NADA may not always match local meteorological droughts or agricultural droughts reported in historical observations.

For the area between 30˚ and 50˚N, covering this study area, validation tests produced median CVRE (calibration period reduction of error calculated by leave-one-out cross-validation) of 0.43, VRSQ (verification period square of the Pearson correlation) of 0.33, VRE (verification period reduction of error) of 0.31, and VCE (verification period coefficient of efficiency) of 0.27 (Cook et al., 2010). However, NADA reconstruction skill varies significantly by location. Among the areas covered by this study, reconstruction skill is highest in New Mexico and North Carolina, and high in most of the southeastern US and Virginia, but reduced in Florida, and poorer in Quebec, the New England coast, and especially Nova Scotia.         (See      complete      validation      statistics      for      each      grid-point      at https://www1.ncdc.noaa.gov/pub/data/paleo/drought/LBDA2010/LBDA_PMDI_JJA_Recons_cal:ver_stats.txt). Other tree ring-based precipitation reconstructions have produced similar results, with similar timing and magnitude of droughts during the early colonial period, for several locations, including New Mexico (Meko et al., 1995; Grissino-Mayer et al,. 2002; Margolis et al., 2011), northern Florida (Harley et al., 2017), Georgia and the Carolinas (Stahle and Cleaveland, 1994), and Virginia (Stahle et al., 1998). Studies of band thickness in a New Mexico speleothem (Asmerom et al., 2013) and $\partial^{18}O$ in a West Virginia speleothem (Hardt et al., 2010) provide some confirmation at lower temporal resolution for major droughts identified in NADA for those regions during the late 16th century and early 17th century, respectively.

This study systematically compared the NADA reconstruction to the information in the documentary record of each expedition identified in section 2.1. Using the visualization tools at http://drought.memphis.edu/NADA/ this study produces: (1) maps of PDSI for each region of travel where definite indications of drought were found in the documentary record; and (2) time series of PDSI variation at the location of specific colonies discussed in the text (section 3.2). Where the quality and quantity of evidence permitted, the routes taken by expeditions and/or the locations of settlements were added to the NADA maps, based on information in White (2017).

## 3. Results

### 3.1. Regions where droughts were not documented

Early colonial expeditions left no first-hand written records clearly indicating droughts in any of the following regions: the California coast, the New England coast, Quebec, or Nova Scotia, apart from an observation of "fine weather all winter [1606-07]" at Port Royal (Nova Scotia). The NADA PDSI reconstruction indicates drought in New England (that is, PDSI between -1 and -2 for most grid-points in the region) during 1603, when an English voyage passed by the Massachusetts coast in summer, and in 1604, when a French colony was founded at the mouth of the St. Croix River in autumn; it indicates drought throughout Nova Scotia (PDSI ≤ -2) in 1606-07, while a French colony was settled on the west coast of the peninsula; and it indicates drought throughout southern California (PDSI between -1 and -2 for most grid-points in the region) during 1542 and 1595, when Spanish expeditions were exploring the coast.

### 3.2 Regions where drought were documented

Early colonial expeditions left first-hand written records indicating droughts in the following regions: the southeastern US, the southwestern US, and Virginia and North Carolina.   Table 1 provides a summary of the type of information concerning drought from these expeditions.

{Table 1 here; caption: Summary of written evidence. A capital letter 'Y' denotes a definite impression of drought in either the winter or summer half-year; a lower-case 'y' denotes an indefinite impression. Letters in **bold** indicate many mutually supporting observations, letters in regular font indicate similar observations in two or three personal sources and/or information contained in an official report, and letters in *italics* indicate a single observer. A letter 'N' indicates information in the

documentary record indicating abundant precipitation or lack of drought. An explanation of sources and types of evidence is provided in section 2.1; details for each expedition are given in sections 3.2.1-3.2.3.}

### 3.2.1. Southeast

The first Spanish colonial expeditions to the Southeast, led by Juan Ponce de León in 1513 and 1521 and Lucas Vázquez de Ayllón in 1526, left no first-hand evidence concerning climatic conditions during their expeditions. In 1528, Pánfilo de Narváez led an overland expedition along the Gulf Coast from roughly Tampa Bay to the mouth of the Mississippi, where its members, travelling on rafts, were swept by a storm to the Gulf Coast of Texas. A few survivors remained in the region as captives of Native Americans until 1534; their testimonies, recorded years later, described cold and famine among the

indigenous populations but left no definite indications of drought during these years. The NADA reconstruction indicates droughts (PDSI ≤-1) in parts of Florida and the Gulf Coast traversed by these expeditions in 1521, 1528-29, and 1532.

During 1539-1543 a much better documented Spanish expedition led by Hernando de Soto landed in northern Florida and travelled throughout the southeastern United States before departing down the Mississippi River. No observers described drought conditions in 1539-40, during which the NADA indicates a positive hydroclimate anomaly. Multiple independent

eye-witness accounts describe a 1541 episode in which Native Americans in present-day Arkansas asked Soto to pray for rain in order to end a drought and save their maize crop. In early 1542, the expedition ventured into eastern Texas, where one witness reported frequent wet weather; late that year, the expedition returned to Arkansas, which another source described as experiencing continued drought (specifically, it had not rained for the past month). The NADA map for 1541 (Figure 1a) indicates a drought (PDSI ≤-1) centered on northeast Arkansas during 1541, while most of the region traversed by Soto that

year enjoyed moist conditions (PDSI >0). The NADA map for 1542 indicates a positive hydroclimate anomaly in east Texas and negative hydroclimate in Arkansas (Figure 1b). The NADA time series for the approximate site of the rain-making ceremony (Figure 1c) indicates that 1541 was the start of a longer drought at this location, as measured by PDSI.

{Figure 1a here; caption: NADA JJA PDSI reconstruction for 1541, Soto expedition}

{Figure 1b here; caption: NADA JJA PDSI reconstruction for 1542, Soto expedition}

{Figure 1c here; caption: NADA JJA PDSI reconstruction time series for 1530-1550 at approximate location of 1541 observed rain-making ceremony}

In late summer 1559, Spanish conquistador Tristán de Luna y Arellano led over 1,500 soldiers and colonists to settle in Pensacola Bay, Florida, where a hurricane soon destroyed most of their ships and supplies. Unable to live off the land and facing famine, the colony retreated inland to central Alabama, while a detachment explored the region of northeast Alabama

and northwestern Georgia, thought to be more promising for settlement. The expedition's breakdown left ample official documentation, including many complaints about the region's environment and climate. These mentioned the frequency and unpredictability of rains—in contrast to those of Mediterranean climates on the same latitude across the Atlantic—but not drought. The NADA composite map of the area in 1559-1560 (Figure 2a) indicates slightly dry conditions (PDSI between 0 and -1) around Pensacola Bay and slightly wet conditions (PDSI between 0 and 1) in northeastern Alabama and northwestern

Georgia, with average conditions in central Alabama, as also indicated in the time series for the approximate location of the colony in that region (Figure 2b).

{Figure 2a here; caption: NADA JJA PDSI reconstruction for 1559-1560, Luna expedition}

{Figure 2b here; caption: NADA JJA PDSI reconstruction time series for 1550-1570, central Alabama}

Between 1562 and 1565, French Huguenots attempted to establish colonies on the coast between present-day South

Carolina and northern Florida. These were over-run in 1565 by Spanish soldiers, who established the permanent colony of St. Augustine, Florida, as well as several outlying posts including St. Elena, which lasted until 1587. The French colonies, as described in several personal accounts by eye-witnesses, suffered from frequent shortages and sometimes famine blamed on poor supplies, shipwrecks, and inability to obtain food from local indigenous communities. The Spanish presence left a much

more complete official record, principally letters from the governor to the viceroy and the correspondence of Catholic missionaries. The former describe droughts that led to failure of wheat and maize crops during 1565-66 in the region between South Carolina and northern Florida; the latter describes Native Americans asking the colony's governor at St. Augustine to pray for rain in 1566 to end a long drought. Official correspondence also indicates that drought led to poor harvests at the St. Augustine colony in 1583, 1588-89, 1591, and 1598-99; for instance, the governor reported that 1588 was a "very great drought" (*muy gran seca*) and maize could not be sown. Food shortages at the colony and surrounding Indian missions were reported for the early 1590s, but these were blamed on late shipments of supplies rather than drought. From 1601 onward, it is possible to calculate maize tithes for most years, and these calculations reveal poor harvests in 1604 and 1609 (Hoffman 2002: 84-88). The NADA composite map for 1565-1568 (Figure 3a) indicates a severe negative PDSI anomaly centered on South Carolina while the NADA time series at St. Augustine for 1560-1580 (Figure 3b) and 1580-1610 (Figure 3c) indicate drought (PDSI ≤-1) 1566-70, 1575, 1583-89, 1591-95, 1598-99, 1604, and 1610.

{Figure 3a here; caption: NADA JJA PDSI reconstruction for 1565-68, French and Spanish Florida}

{Figure 3b here; caption: NADA JJAs reconstruction time series for 1560-1580, St. Augustine}

{Figure 3c here; caption: NADA JJA reconstruction time series for 1580-1610, St. Augustine}

### 3.2.2 Southwest

During 1534-1535, the four survivors of the Pánfilo de Narváez expedition (see section 3.2.1) ventured south into today's Tamaulipas (Mexico) then northwest near the Rio Grande valley. Their testimonies describe dry conditions in the Sierra Madre Oriental but only a single definite indication of drought: survivor Cabeza de Vaca reported that Indians in west Texas in late summer 1535 asked them to pray for rain because it had not rained enough to plant maize during the past year. A comparison with the NADA map of PDSI for that year (Figure 4), indicates average to dry conditions in that region in 1535.

{Figure 4 here; caption: NADA JJA PDSI reconstruction for 1535, Cabeza de Vaca expedition}

During 1540 and 1541, a large expedition from Mexico led by governor Francisco Vázquez de Coronado occupied first the Zuni pueblos of today's western New Mexico, then various Puebloan towns of the central Rio Grande valley. Of the many surviving official and personal accounts of the expedition, several mentioned heavy winter snows and none mentioned drought. The NADA composite map for New Mexico 1540-1541 (Figure 5a) shows exceptionally moist conditions (PDSI between 2 and 3), as does the time series for the approximate location of the Spanish base camp, near present-day Bernalillo (Figure 5b).

{Figure 5a here; caption: NADA JJA PDSI reconstruction for 1540-1541, New Mexico}

{Figure 5b here; caption: NADA JJA reconstruction time series for 1530-1550, at the approximate location of Coronado's base, near present Bernalillo, NM}

During 1581, 1583, and 1590-1591, three small unauthorized expeditions entered New Mexico from New Spain. Each travelled through the territory of the present-day state for only a few weeks or months, mainly along the Rio Grande, leaving no enduring colony; and each produced only two or three written testimonies by eye-witnesses. A witness of the 1581 expedition described a "climate like that of Castile": a possible indication of deficient summer rainstorms, since unlike Castile, most of New Mexico usually receives more precipitation in summer than winter. He also described a Puebloan rain-making ceremony, although it is not clear whether this was a normal seasonal ritual or a specific response to drought. A witness from the 1583 expedition described heavy snows early in the year but reported Puebloans refusing to share maize due to a "lack of rain" and poor harvest that summer. The 1590-1591 expedition travelled mainly during the winter months and left testimonies of exceptional cold and heavy snows but not drought. The NADA reconstructions indicate near average conditions in 1581 (Figure 6a) and negative PDSI anomalies across most of the present-day state in 1583 (Figure 6b) and in 1591 (Figure 6c).

{Figure 6a here; caption: NADA JJA PDSI reconstruction for 1581, New Mexico}

{Figure 6b here; caption: NADA JJA PDSI reconstruction for 1583, New Mexico}

{Figure 6c here; caption: NADA JJA PDSI reconstruction for 1591, New Mexico}

A 1598 invasion led by Juan de Oñate occupied present-day New Mexico and subjugated the Pueblos of the region. In 1601, the expedition nearly collapsed when a large number of officers and settlers defected and Catholic missionaries protested the abuse of Pueblos. Narratives of Oñate's campaign, as well as disputes arising from the colony's near collapse, left a large body of personal and official evidence concerning conditions in New Mexico. The earliest indication of a drought in the documentary record occurs in a description of Pueblos asking colonists to pray for rain in 1598; however, this episode appears only in one contemporary source, which contains some fictionalized material. Colonists during 1599 and 1600 compared the climate in New Mexico to that of Castile, with predominately winter precipitation, possibly indicating deficient summer rainstorms. Colonists and missionaries left abundant official testimonies and personal letters describing extreme summer drought during 1600 and 1601, along with descriptions of starvation when the Pueblo maize crop failed and livestock died for want of pasture. However, descriptions also indicate that the winter of 1600-01 was exceptionally snowy and icy and that the colony's crop of irrigated winter wheat yielded well. For the years 1602-10, there is no comparable indication of drought in the documentary record. The NADA composite map for New Mexico during 1598-1601 (Figure 7a) shows a significant negative PDSI anomaly, particularly in the central and northern parts of the present-day state, where the Oñate expedition settled. The NADA times series for the location of Oñate's first colony at present Okhay Owingeh pueblo (Figure 7b) indicates negative PDSI anomalies in 1598 and 1600-1601, although not as significant as those of the "megadrought" during the 1580s.

{Figure 7a here; caption: NADA JJA PDSI reconstruction for 1598-1601, New Mexico}

{Figure 7b here; caption: NADA JJA PDSI reconstruction time series for 1580-1610, at Oñate's first colony, present Okhay Owingeh, New Mexico}

### 3.2.3 Virginia and North Carolina

In 1570, a small group of Spanish Jesuit missionaries attempted to establish an outpost along the lower James River in present-day Virginia. In the scarce documentation that survives from the expedition, they reported that the land had been "punished with six years of sterility and death" and was "very parched (*muy agostada*)" such that both maize and wild plants had died. Native Americans massacred the priests in early 1571. The NADA composite map for 1565-1570 indicates a significant negative PDSI anomaly for the region of the outpost (Figure 8a), and the time series of PDSI variation for the approximate location (Figure 8b) indicates persistent if not always severe drought since 1562, with the worst year (PDSI=-3) occurring in 1566.

{Figure 8a here; caption: NADA JJA PDSI reconstruction composite map for 1565-1570, Virginia}

{Figure 8b here; caption: NADA JJA PDSI reconstruction time series for 1560-1580, approximate location of Ajacán}

Between 1585 and 1587, English investors led by Sir Walter Raleigh made several attempts to colonize Roanoke Island in the Carolina Outer Banks, before losing all contact with the final "lost colony." A settler described how in 1586 the Indians' corn "began to whither by reason of a drought which happened extraordinarily" and reported being asked to pray for rain to end that drought. During the following year, the colony was unable to obtain maize from indigenous communities, who reported scarcity, and the colonists therefore suffered famine. The NADA composite map for 1585-1588 (Figure 9) shows a major drought throughout eastern North Carolina, although not specifically for Roanoke Island, which is not covered by the atlas.

{Figure 9 here; caption: NADA JJA PDSI reconstruction for 1585-1588, eastern North Carolina}

The English colony at Jamestown, Virginia suffered through multiple environmental disasters and conflict with Native Americans from its founding in 1607 through 1610, when it was rescued by a large infusion of new settlers and supplies. Several contemporary personal letters, as well as pamphlets and histories later written by eye witnesses, left indications of exceptional summer drought throughout these first years at the colony. In summer and autumn of 1607, saltwater apparently

intruded into the James River as far as Jamestown, where the water was described as "at flood very salt, at low tide full of slime and filth"; this unusual occurrence may have caused salt poisoning at the colony (Earle, 1979). Settlers' summer grain crops failed repeatedly for unspecified reasons. Multiple colonists also recorded complaints from Indians in villages throughout the region during 1608-10 that they had no maize to share with colonists or even for their own consumption. Two

sources reported that in 1608 or 1609 a leader of a nearby indigenous community asked a colonist to "pray to his god for rain, for his [own] god would not send him any." During the summer of 1610, sturgeon did not swim up the James River as usual, which may indicate high salinity in that waterway. The NADA composite map for 1606-1610 (Figure 10a) indicates a moderate drought over southeastern Virginia. The NADA time series of PDSI variation for 1600-1620 at the location of Jamestown (Figure 10b) indicates a negative PDSI anomaly each summer during 1606-1612, with a severe drought (PDSI=-

3) in 1610, making this span of years Virginia's longest continuous drought of the past seven centuries, as reconstructed from local tree ring width (Stahle et al., 1998).

{Figure 10a here; caption: NADA JJA PDSI reconstruction for 1606-1610, eastern Virginia and North Carolina}

{Figure 10b here; caption: NADA JJA PDSI reconstruction time series for 1600-1620, Jamestown, VA}

**3.3 Synthesis**

Table 2 presents a synthesis of the results described in sections 3.1 and 3.2. It indicates the years and regions where documented expeditions were present, and one of the following results: (1) droughts identified in both the documentary record and NADA reconstruction; (2) droughts identified in neither; (3) droughts identified in the NADA reconstruction but not the documentary record; (4) instances of partial or possible agreement between the documentary record and NADA, including

instances where the documentary record identifies the impacts of a long drought but does not specify each year of its occurrence; (5) no first-hand written evidence concerning climatic conditions.

{Insert Table 2 here; Caption: Synthesis of comparison between drought information in the documentary record of early North American colonial expeditions and the NADA PDSI reconstruction. Green indicates droughts identified in both the documentary record and NADA reconstruction; blue indicates droughts identified in neither; red indicates droughts identified

in the NADA reconstruction but not the documentary record; yellow indicates instances of partial or possible agreement between the documentary record and NADA, including instances where the documentary record identifies the impacts of a long drought but does not specify each year of its occurrence; dark grey indicates that there is no first-hand written evidence concerning climatic conditions}

**4. Discussion**

As indicated in Table 2, concurrence between the documentary record and the NADA PDSI reconstruction on the timing of droughts is not complete but clearly better than would be expected by chance. The documentary record produced no false positives vis-à-vis the NADA reconstruction—that is, the documentary record never provided a definite indication of a drought where the NADA indicated a positive hydroclimate anomaly (PDSI >0)—but observers did not leave written indications of all

droughts (PDSI ≤-1) appearing in the NADA reconstruction. In certain cases, the documentary records and NADA reconstruction show a remarkable correspondence. Records of the Soto expedition identified drought in precisely the regions and years where it appears in the NADA maps and not elsewhere; and records of the first expeditions to Virginia and North Carolina each indicated droughts in agreement with the tree ring-based reconstruction. In other cases, such as New England and Canadian expeditions, there is little correspondence between the two.

40          The results of the comparison point to three main patterns. First. the documentary information and tree ring-based reconstruction frequently disagree where the NADA spatial coverage or reconstruction skill is poorest, including the New England coast and Nova Scotia. Second, during multi-year droughts in Florida, observers did not necessarily record the occurrence or impacts of the drought each year but only at the beginning or end of the drought. A study comparing

documentary information and tree-ring reconstructed hydroclimate variability has found a similar pattern for medieval Hungary (Kiss, 2017). Third, concurrence between the NADA reconstruction and the documentary record—and in this case, probably the reliability of climatic information in early colonial written sources—is highest when and where colonial observers were able to exchange information with local indigenous populations, thanks to presence of at least one translator and cultural intermediary, whether a Native American who had lived among Europeans or vice versa. This was the case for expeditions and colonies in the southeast and Florida in 1539-41 and 1565-1610; the southwest in 1534-35 and 1598-1610; and North Carolina and Virginia in 1570, 1585-87, and 1608-10. This pattern suggests that early colonial observers obtained the most reliable indications of climatic variability and extremes from local populations and their impacts and reactions.

The results suggest no obvious differences in the reliability of different source types (official, personal, or ecclesiastical) or nationalities (Spanish, English, or French). In general, colonial observers were most likely to record droughts that had a direct impact on their own livelihoods. However, in this small sample, reports of rain prayers and indigenous societies afflicted by drought also turned out to be reliable indicators, even when there were no other mentions of drought.

Written descriptions of drought nearly all concern the spring and summer, probably because spring and summer rains were vital to North America's staple crop, maize. Descriptions of summer droughts in New Mexico, in particular, were often combined with descriptions of snowy winters, particularly during 1599-1601, suggesting that the reconstructed negative PDSI anomaly for these years reflected a summer precipitation deficit. However, a tree ring-based reconstruction attempting to distinguish winter and summer precipitation signals using separate early- and latewood measurements has concluded that New Mexico's "megadrought" of 1580-1600 affected all seasons (Stahle et al., 2009). Most years of drought identified in the documentary record and NADA reconstruction were also described by observers as unusually cold (White, 2017). This coincidence suggests that deficits in growing season soil moisture appearing in the NADA reconstruction as negative PDSI anomalies should have been due to reduced precipitation rather than increased evapotranspiration, unless the cold had a confounding effect on tree-ring growth in local samples.

## 5. Conclusion

This study finds broad agreement between the evidence in written records and the tree-ring based NADA concerning the occurrence and severity of droughts in North America during 1510-1610CE. In years and locations where substantial written observations were made and the NADA reconstruction skill is high, the resulting documentary records did not indicate drought where the NADA reconstruction indicated a positive hydroclimate anomaly (PDSI >0) and usually did indicate drought where the NADA reconstructed indicated a negative hydroclimate anomaly (PDSI ≤-1). This convergence of evidence suggests that early colonial records may provide useful sources of climate and weather information. The advantages of early colonial sources for historical climatology, including their authors' interest in environmental factors and vulnerability to environmental extremes, appear to balance some of their disadvantages, such as their authors' unfamiliarity with local climates. This finding should guide the use of early colonial records to investigate climatic vulnerabilities and adaptations of indigenous peoples before their societies were transformed by colonialism and introduced pathogens (e.g., White, 2014; Wickman, 2018). Hence, this informs studies of proto-historic climate impacts based on the archaeological record and paleoclimate proxies (e.g., Anderson, 1994; Bird et al., 2017). Moreover, this study suggests that early colonial observers were capable of accurately recording features of the environments they encountered, thus offering some support for the use of these records in proto-historical and early colonial environmental history (e.g., Cronon, 1983).

The results also suggest that the NADA reconstruction could be precise enough to help examine historical drought conditions and impacts at a local human scale, at least for periods and locations where the reconstruction skill of the drought atlas is particularly good (e.g., the southwestern US during the past five centuries). In this respect, the NADA's exceptionally high density of samples covering much of the continent may enable types of detailed climate history studies not yet possible for other parts of the world. By contrast, the Old World Drought Atlas—based on 106 tree-ring chronologies and with lower

average reconstruction skill for each grid point (median CVRE=0.271; VRSQ=0.198; VRE=0.161; VCE=0.146)—does not reproduce many local hydroclimate events described in written records (e.g., Collet, 2018: 74) and may be more suited for identifying extensive multi-year events (Cook et al., 2015; Kiss, 2017).

This general concurrence between the early colonial documentary record and the NADA reconstruction does not mean that such incomplete documentary records are themselves capable of producing drought reconstructions. However, the result does suggest how incomplete documentary records demonstrating concurrence with high-resolution proxy-based reconstructions might be combined with other sources in order to arrive at better estimates for past climate conditions. One might estimate from comparisons with proxy-based reconstructions that a type of documentary record has a high probability of containing certain observations whenever a climate parameter has certain values (p(O|H)) and a low probability of containing those observations whenever the parameter has different values (p(O|~H)). Thus by using the ratio p(O|H)/p(O), the presence or absence of those observations in the records for a given time and place might be used to derive posterior probability estimates for those parameter values at that time and place from priors provided by proxy-based reconstructions or models.

Finally, this study has implications for the role of climate in early North American colonial history. Although early colonial observers did not record climate conditions perfectly, neither apparently did they exaggerate the occurrence of drought. Therefore, historians should take seriously the descriptions in several early expeditions and colonies of drought-driven food shortages, famines, and resulting conflict with indigenous communities over scarce resources. This study generally supports previous research findings that drought substantially altered the outcome of several expeditions and therefore the outcome of competition between the Spanish, French, and English empires for North American territory. However, the results of this study may be even more significant for assessing the role of non-drought climatic factors in early colonial history. Descriptions of drought are less common in early colonial North American records than descriptions of destructive storms and exceptional winter cold, which were reported during many expeditions and may have had an even greater impact on their outcomes (e.g., White, 2014). Since storms and winter temperature are more difficult than drought to reconstruct from proxies in natural archives, it is especially important to establish the reliability and objectivity of information in the documentary record in order to assess the role of those climate factors in early colonial history.

**The author reports no conflicts of interest. The author thanks Bill Keegan, for assistance in the preparation of maps, and the PAGES-CRIAS working group.**

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

**Table 1** Summary of written evidence. A capital letter 'Y' denotes a definite impression of drought in either the winter or summer half-year; a lower-case 'y' denotes an indefinite impression. Letters in **bold** indicate many mutually supporting observations, letters in regular font indicate similar observations in two or three personal sources and/or information contained in an official report, and letters in *italics* indicate a single observer. A letter 'N' indicates information in the documentary record indicating abundant precipitation or lack of drought. An explanation of sources and types of evidence is provided in section 2.1; details for each expedition are given in section 3.2.

| Region | Year | Location | Phenology | Description | Impact | Rain Prayer |
|---|---|---|---|---|---|---|
| **Southeast** | 1539 | Florida | | | | |
| | 1540 | interior southeast | | **n** | | |
| | 1541 | Arkansas | | | Y | **Y** |
| | 1542 | Arkansas/E Texas | | *y* | | |
| | 1559 | Mississippi | | | | |
| | 1560 | Mississippi | | **n** | | |
| | 1564 | Florida | | | y | |
| | 1565 | S Carolina/Florida | | | y | |
| | 1566 | Florida | | y | Y | Y |
| | 1583 | Florida | | y | Y | |
| | 1588 | Florida | | y | Y | |
| | 1589 | Florida | | | Y | |
| | 1591 | Florida | | | Y | |
| | 1598 | Florida | | | **Y** | |
| | 1599 | Florida | | y | **Y** | |
| **Virginia/North Carolina** | 1570 | Virginia | | *y* | Y | |
| | 1585 | N Carolina | | | | |
| | 1586 | N Carolina | | *y* | Y | *Y* |
| | 1587 | N Carolina | | | y | |
| | 1607 | Virginia | *Y* | | y | |
| | 1608 | Virginia | | y | Y | Y |
| | 1609 | Virginia | | | Y | |
| | 1610 | Virginia | y | | **Y** | |
| **Southwest** | 1535 | W Texas | | | Y | *Y* |
| | 1540 | New Mexico | | n | | |
| | 1541 | New Mexico | | n | | |
| | 1581 | New Mexico | | *y* | | *y* |
| | 1583 | New Mexico | | *y* | Y | |
| | 1590 | New Mexico | | | | |
| | 1598 | New Mexico | | y | | *Y* |
| | 1599 | New Mexico | | y | | |
| | 1600 | New Mexico | | **Y** | **Y** | |
| | 1601 | New Mexico | | **Y** | **Y** | |

**Table 2** Synthesis of comparison between drought information in the documentary record of early North American colonial expeditions and the NADA PDSI reconstruction. Green indicates droughts identified in both the documentary record and NADA reconstruction; blue indicates droughts identified in neither; red indicates droughts identified in the NADA reconstruction but not the documentary record; yellow indicates instances of partial or possible agreement between the documentary record and NADA, including instances where the documentary record identifies the impacts of a long drought but does not specify each year of its occurrence; dark grey indicates that there is no first-hand written evidence concerning climatic conditions

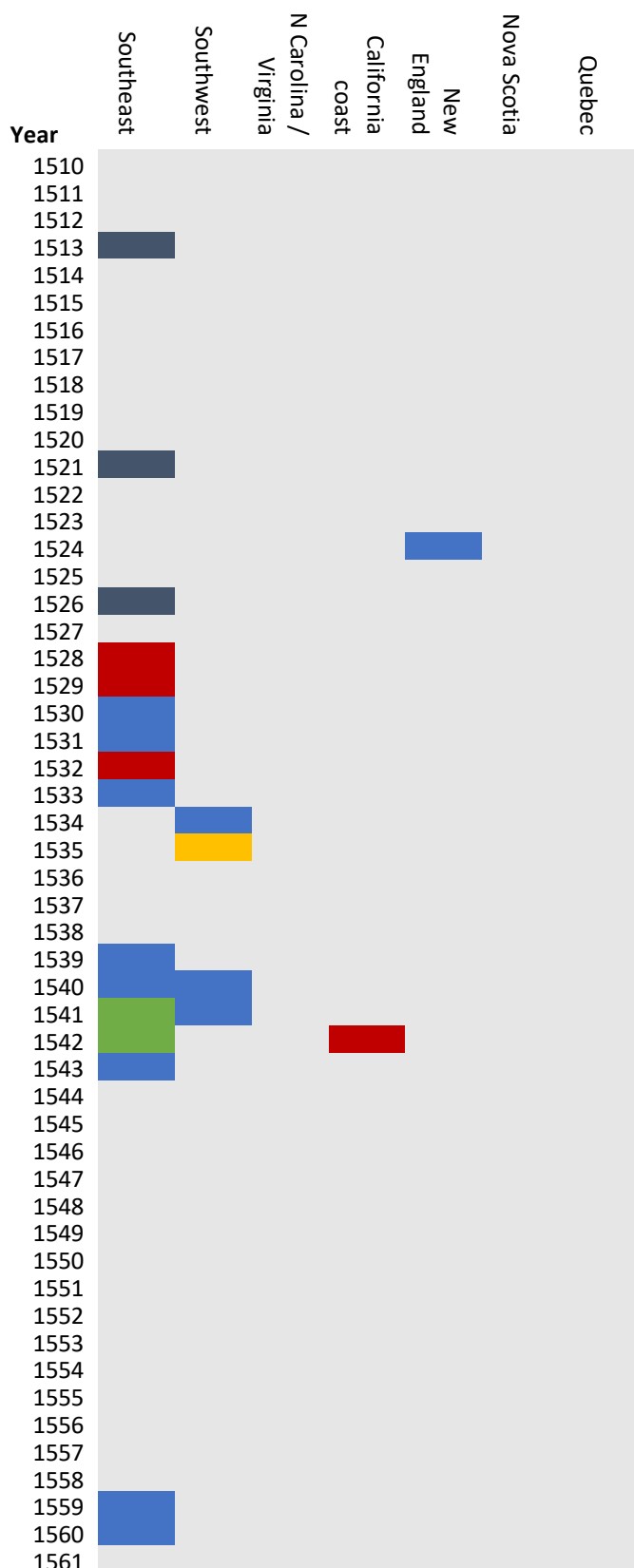

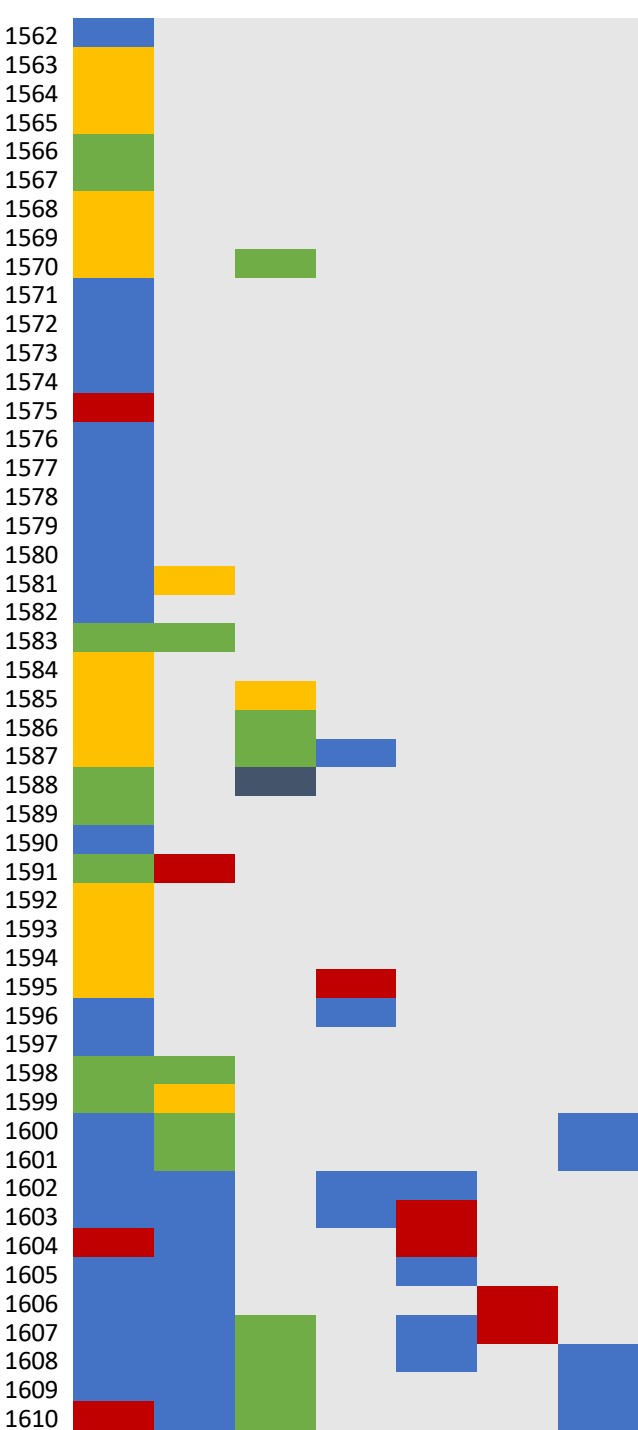

**Figure 1a** NADA JJA PDSI reconstruction for 1541, with approximate route of Soto expedition

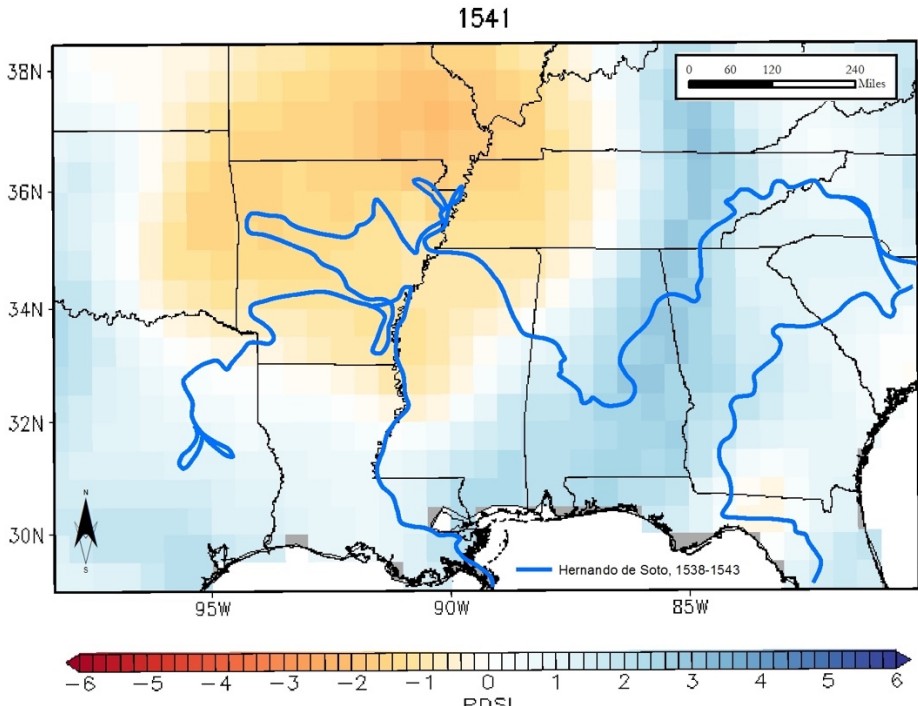

**Figure 1b** NADA JJA PDSI reconstruction for 1542, with approximate route of Soto expedition

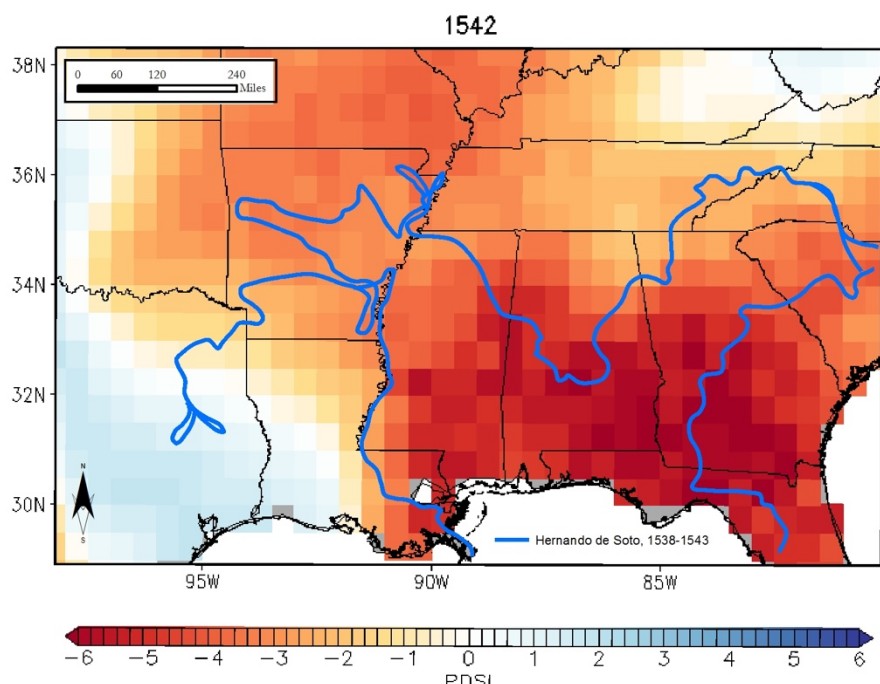

**Figure 1c** NADA JJA PDSI reconstruction time series for 1530-1550 at approximate location of 1541 observed rain-making ceremony

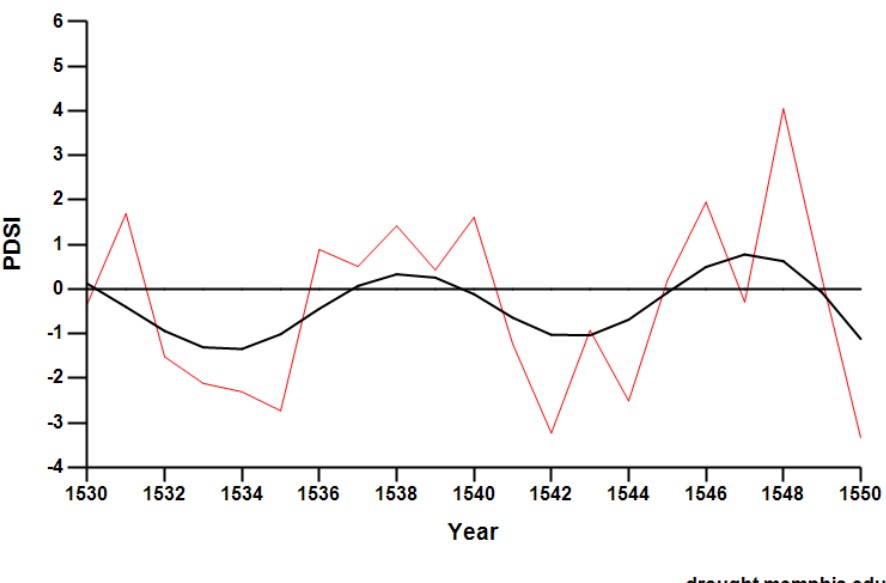

**Figure 2a** NADA JJA PDSI reconstruction for 1559-1560, with approximate route and location of colony

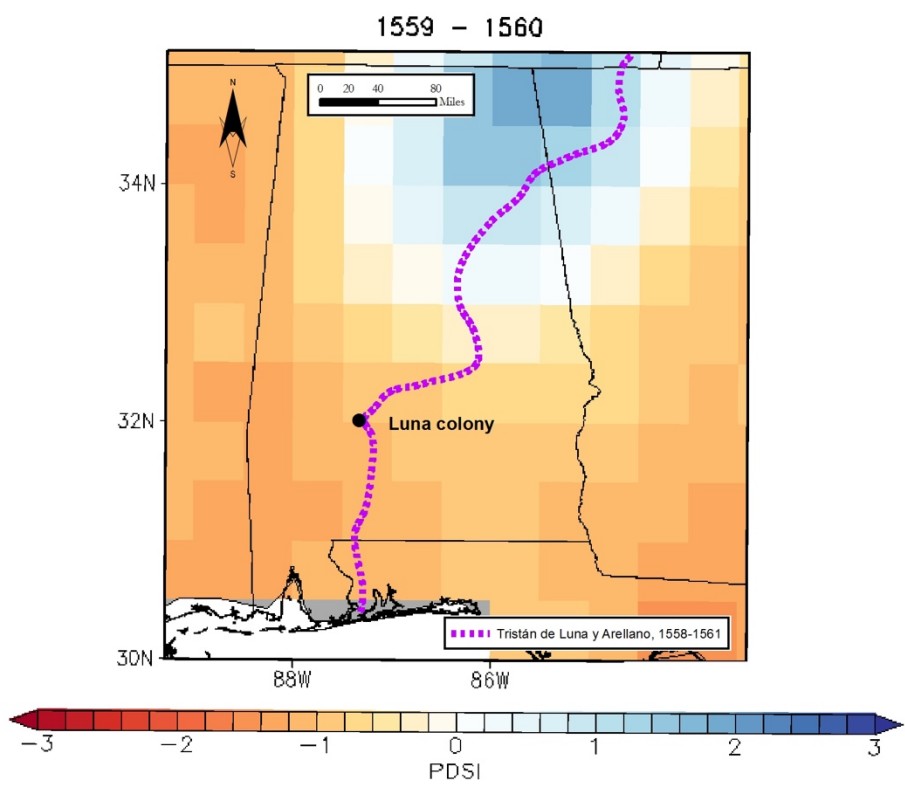

**Figure 2b** NADA JJA PDSI reconstruction time series for 1550-1570, central Alabama

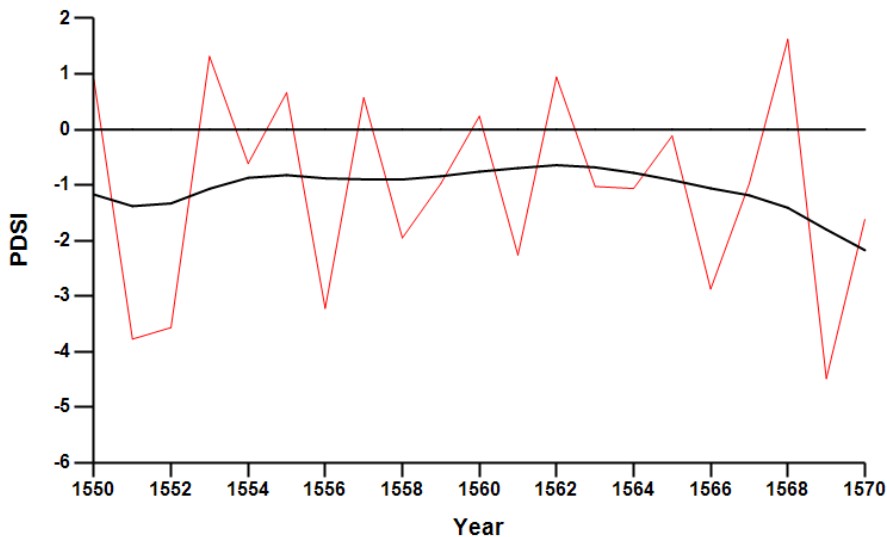

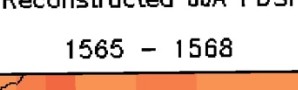

**Figure 3a** NADA JJA PDSI reconstruction for 1565-68, French and Spanish Florida

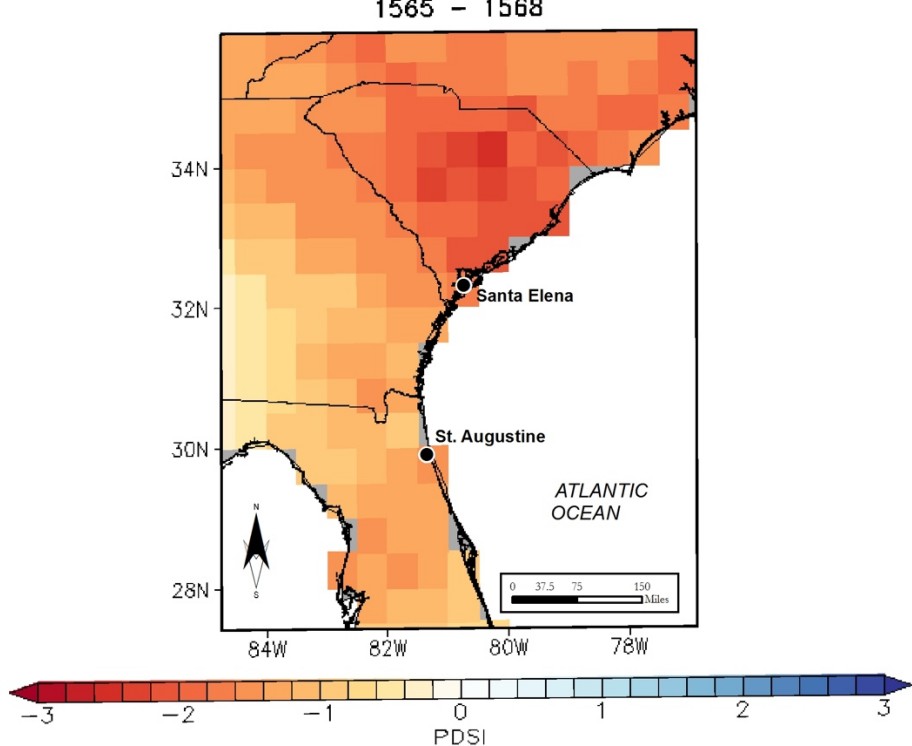

**Figure 3b** NADA JJAs reconstruction time series for 1560-1580, St. Augustine

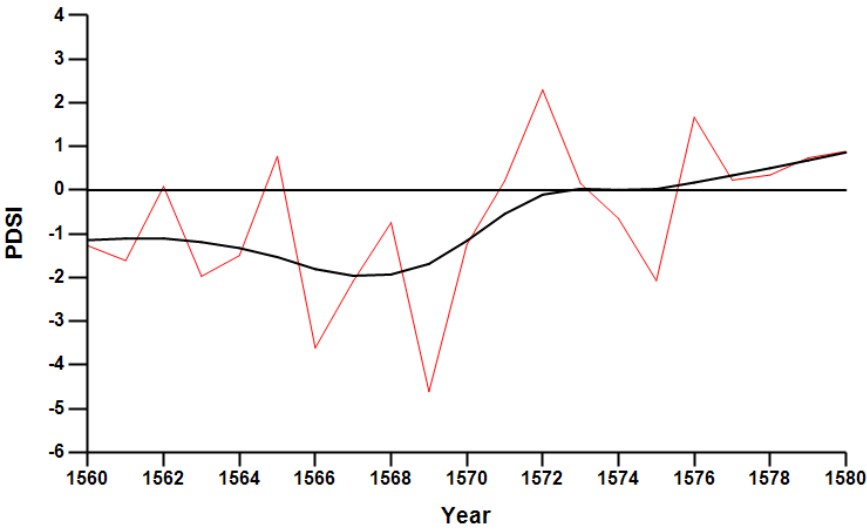

**Figure 3c** NADA JJA reconstruction time series for 1580-1610, St. Augustine

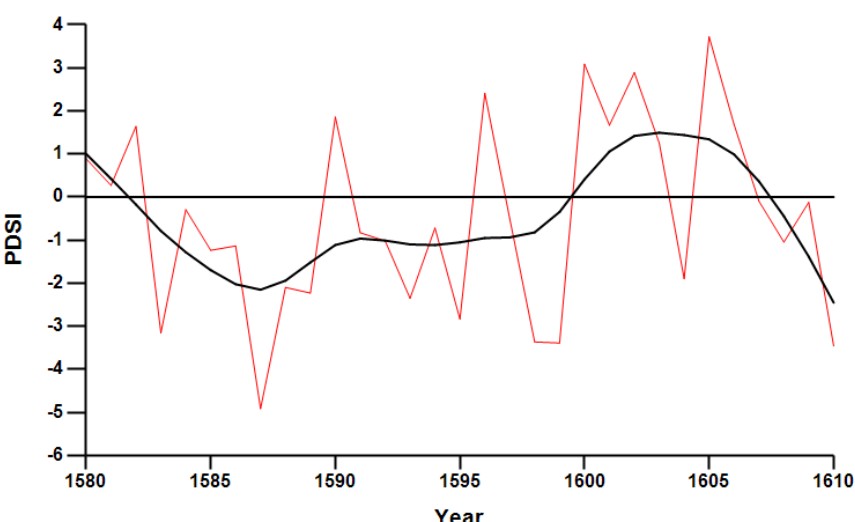

**Figure 4** NADA JJA PDSI reconstruction for 1535, Cabeza de Vaca expedition, with approximate route and location of rain prayer

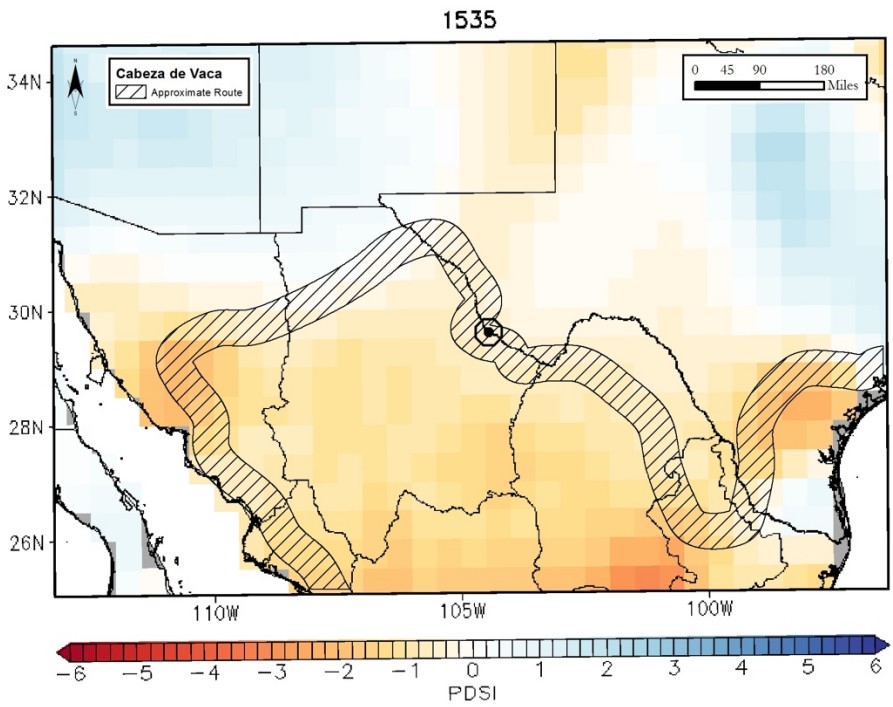

**Figure 5a** NADA JJA PDSI reconstruction for 1540-1541, New Mexico, Coronado Expedition

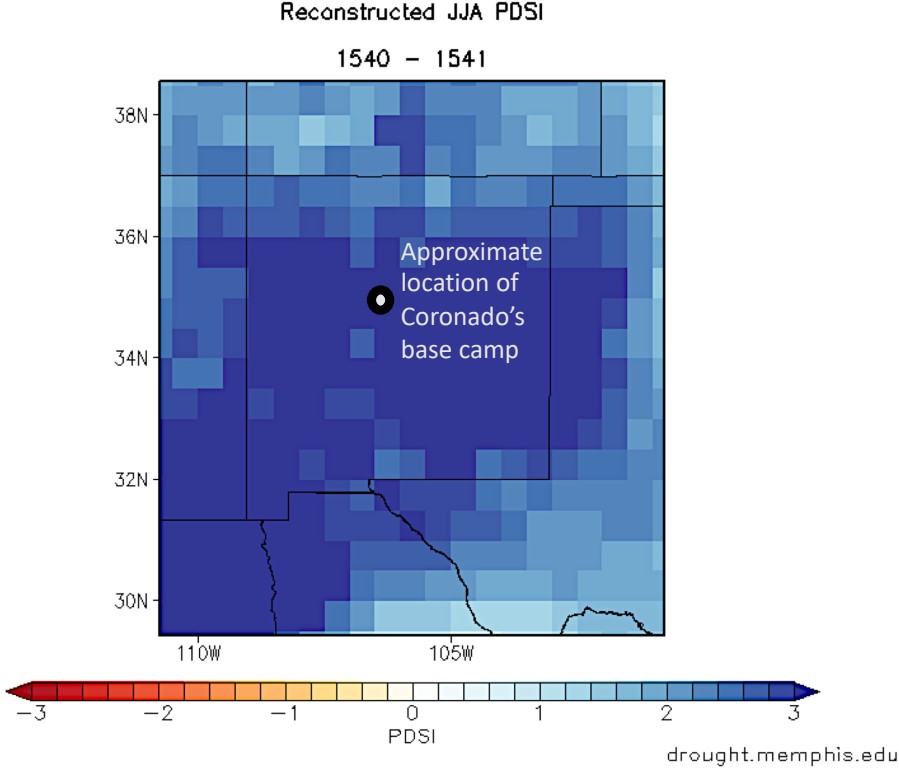

**Figure 5b** NADA JJA reconstruction time series for 1530-1550, at the approximate location of Coronado's base, near present Bernalillo, NM

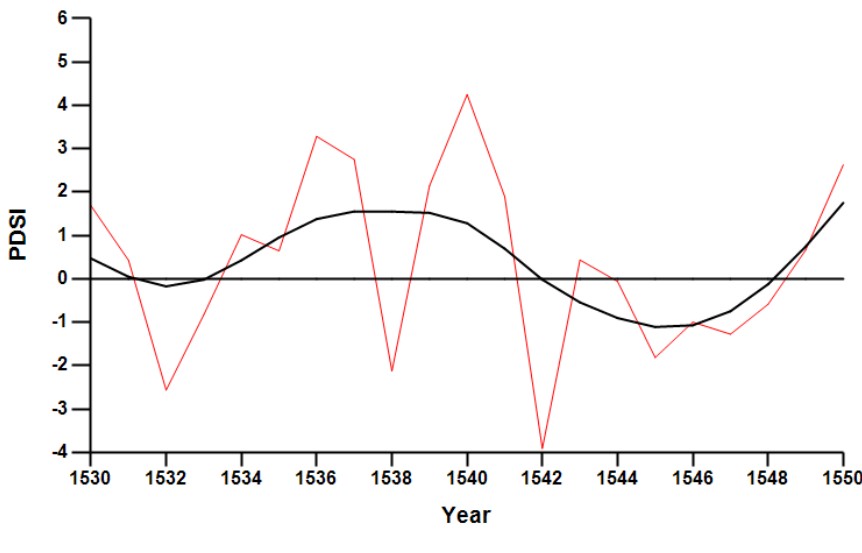

**Figure 6a** NADA JJA PDSI reconstruction for 1581, New Mexico

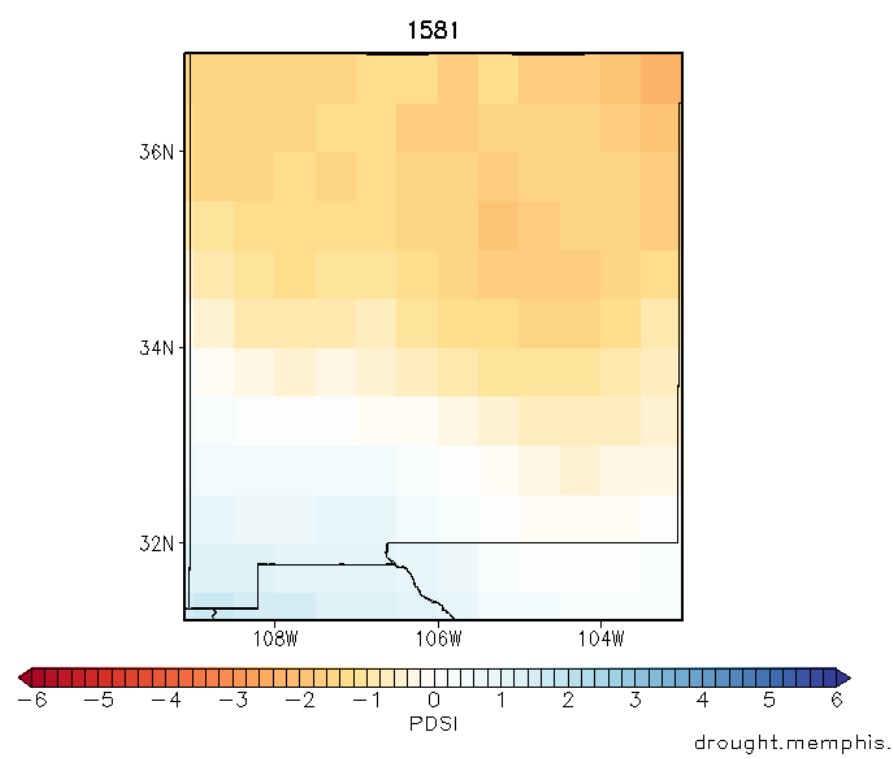

**Figure 6b** NADA JJA PDSI reconstruction for 1583, New Mexico

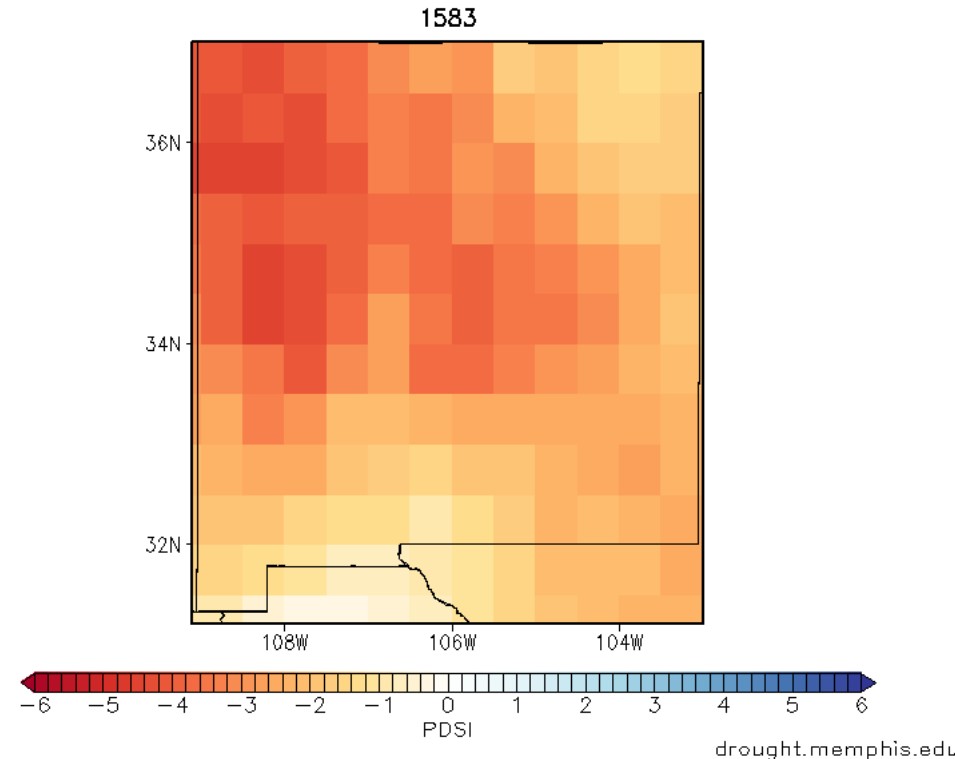

**Figure 6c** NADA JJA PDSI reconstruction for 1591, New Mexico

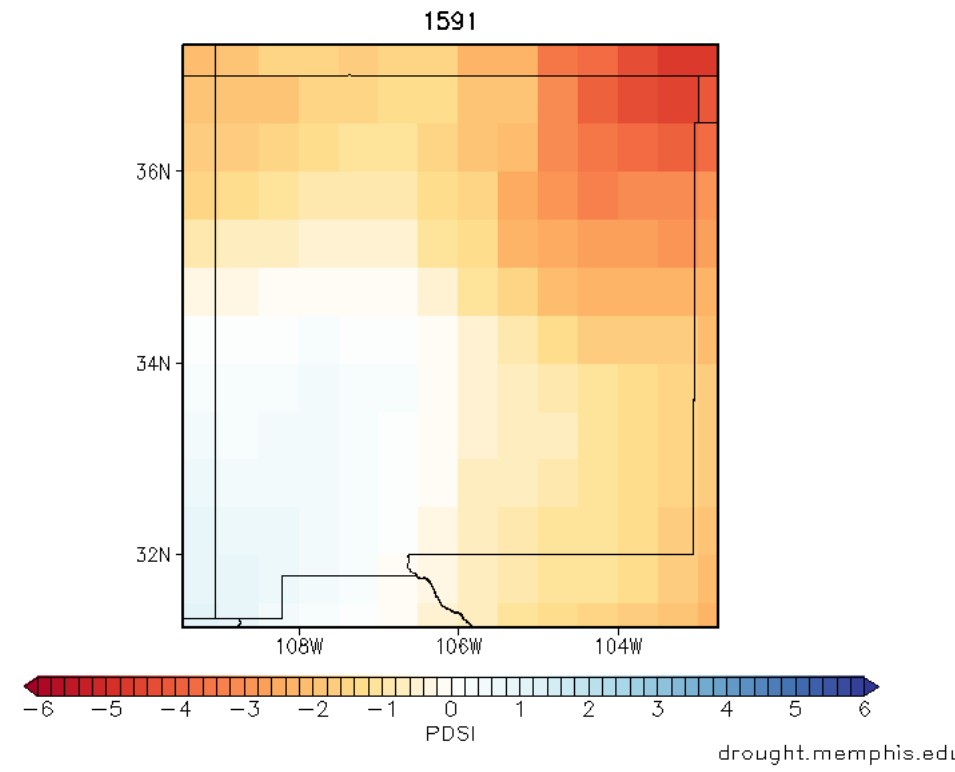

**Figure 7a** NADA JJA PDSI reconstruction for 1598-1601, New Mexico

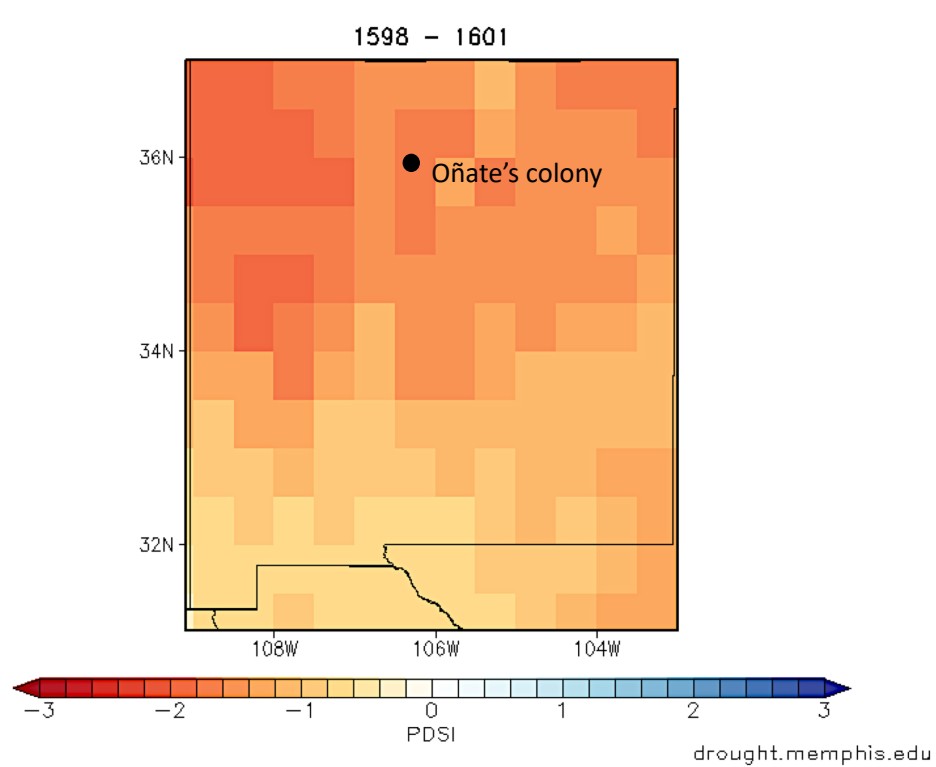

**Figure 7b** NADA JJA PDSI reconstruction time series for 1580-1610, at Oñate's first colony, present Okhay Owingeh, New Mexico

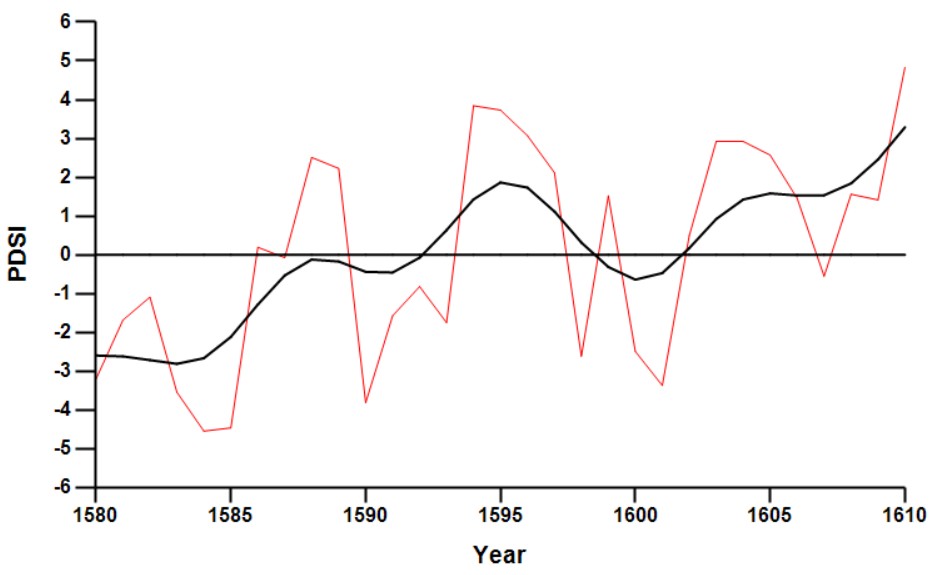

**Figure 8a** NADA JJA PDSI reconstruction for 1565-1570, Virginia

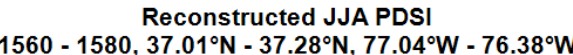

**Figure 8b** NADA JJA PDSI reconstruction time series for 1560-1580, approximate location of Ajacán

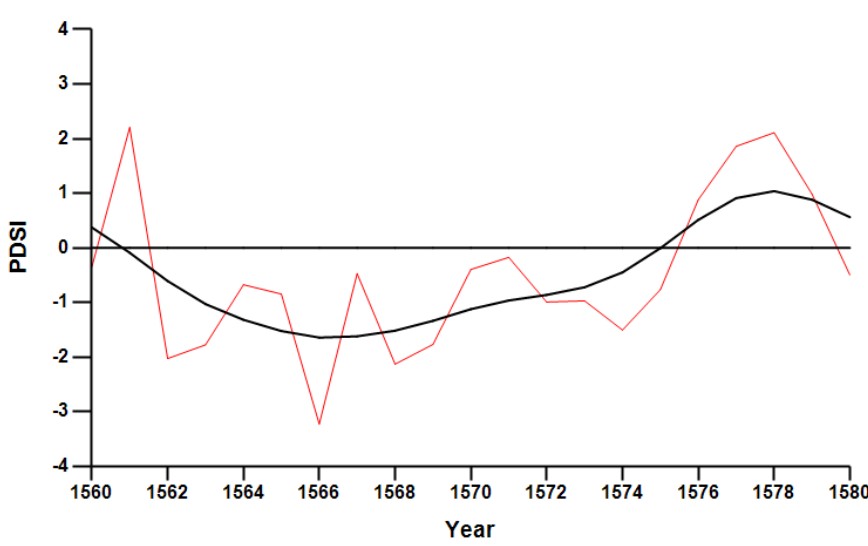

**Figure 9** NADA JJA PDSI reconstruction for 1585-1588, eastern North Carolina

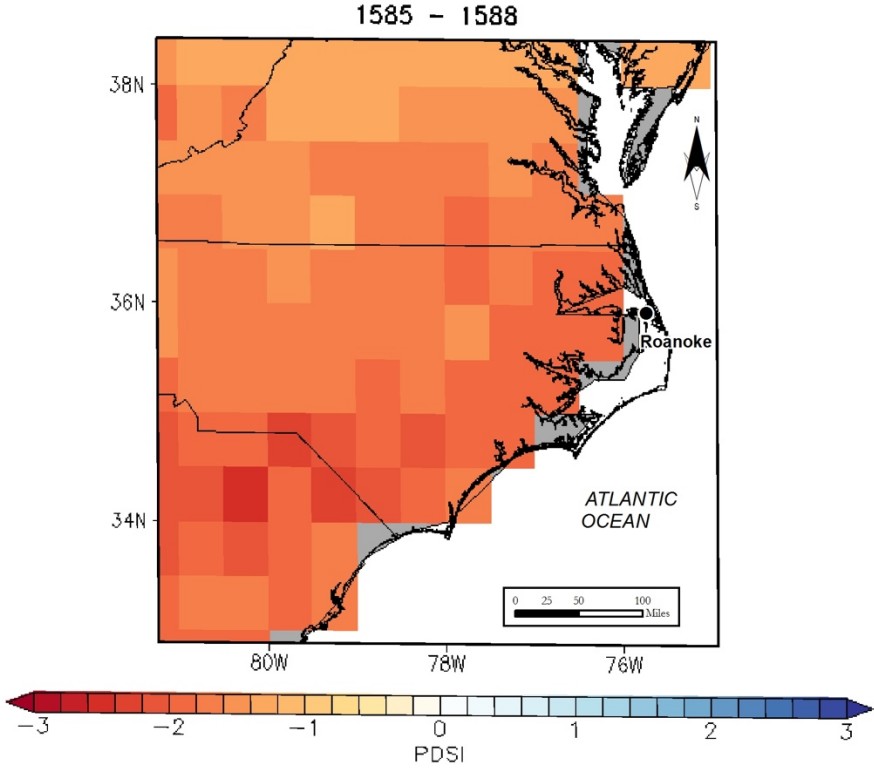

**Figure 10a** NADA JJA PDSI reconstruction for 1606-1610, eastern Virginia and North Carolina

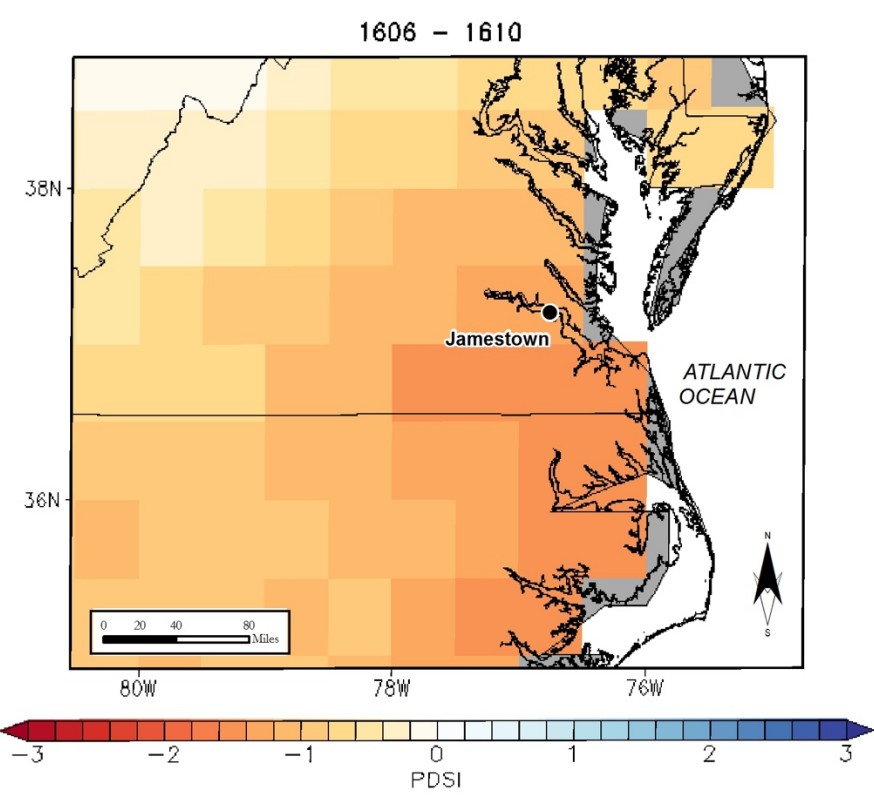

**Figure 10b** NADA JJA PDSI reconstruction time series for 1600-1620, Jamestown, VA

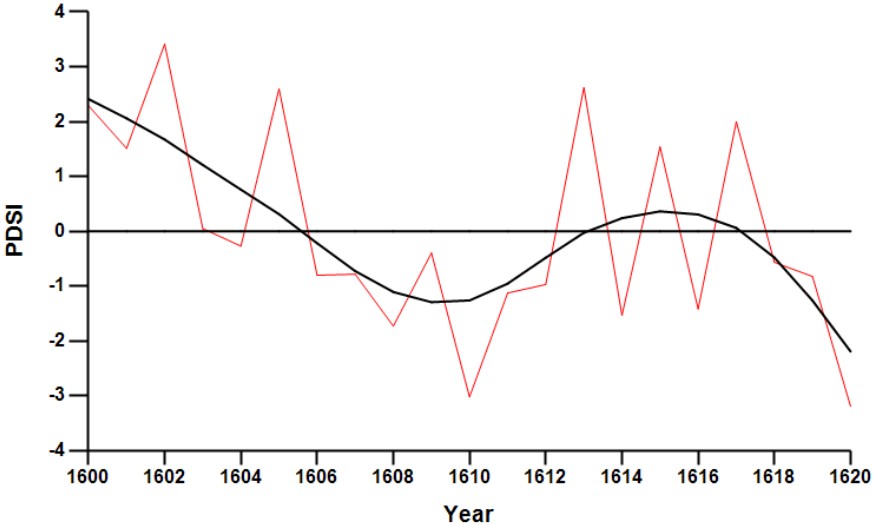