# Peer review of "A Comparison of Drought Information in Early North American Colonial Documentary Records and a High-Resolution Tree Ring-Based Reconstruction"

_Climate of the Past, 2019_

## Referee Comment (RC1) · Anonymous Referee #1 · 11 Feb 2019

Referee comments on: Drought during early European exploration and colonization of North America, 1500-1610CE: A comparison of evidence from the archives of societies and the archives of nature

This paper takes an interesting and somewhat new and refreshing approach to historical weather/climate studies by comparing so called 'archives of society' (written historical documentary accounts) with those of 'nature' (here mostly tree rings). A great idea I think. The study takes a risk by attempting to establish climatic conditions based on documentary evidence, during what would have been the very beginnings of colonial conquest into North America (16th C). This at a time when documentary evidence would have been limited for any given place over any given time (certainly when one compares with the data density for a similar period in a European context).

Overall, the paper is stylistically and typologically excellently written and well organized. The abstract suitably covers what it should and is an accurate representation of the paper. I like the introduction (probably the best part of the paper for me) – it is well structured and introduces the topic in a succinct, clear and critical manner. Many of the challenges are highlighted in the introductory sections of the paper. However, I think some critical challenges are not adequately addressed, and so I will elaborate on some major challenges/concerns that I feel need to be addressed to make this work publishable.

Major concerns: 1. The paper does not define 'droughts' in the North American context. . . ...and doing so for such a vast region is challenging as 'experiencing dry conditions' in specific areas may not necessarily imply drought, especially if it is during a naturally dry season, or in a dry region. Defining drought is also relative to the individuals' past experience of climate (depending where they came from) – so someone who is accustomed to semi-arid conditions is less likely to identify 'drought' conditions as opposed to someone who is accustomed to a climate of all year plentiful rain. Such context must be taken into account in all instances, which I do not see much of in the case study examples presented in this paper. Then of course there are the different types of drought such as hydrological, climatological, agricultural etc and these differ too, yet the paper is unable to differentiate between these. 2. My second major concern is that in the examples presented there is very limited 'societal/historical written evidence' presented to support drought conditions. In most cases there are only one or two lines of evidence and this is surely insufficient, especially given the nature of some of this evidence. 3. This brings me to the third major concern regarding types of evidence used. Famines are of course not necessarily an indication of drought. As the author correctly implies, some of these famines may be due to severe cold, snow, storms,
social disruptions etc. But they may also be due to poor farming practices, poor decisions made with regards planting time or most suitable crops, as also pests that might destroy crops. Praying for rain may also not necessarily imply a drought. Prayer might be asked for if the rains may be delayed, or there may be a mid- summer dry spell etc. . .but if one were to look at the season as a whole it may not have been a drought season. Poor harvests and crop failures may also not imply drought, for some of the same reasons already mentioned above. 4. A tough one here, but classifying a season or a year as a drought season or drought year would surely require one to have some sort of bench-mark to compare against (i.e. with other years). An important question for the author to clarify in this regard is whether, based on the documentary evidence, one is able to say whether a season or year is far enough below the 'normal' to define it as a 'drought'. Or, does this paper simply take dry conditions (irrespective of whether it is below normal, normal, or above normal in rainfall/moisture) which affect society, as meaning it is a 'drought'. Better clarity on all this is required. 5. Are there not a wider variety of evidence types that might be discussed in each case study? For instance, reports of grass being dry or sparse, fires, rivers dried up or far below the normal level, death of natural vegetation due directly to drought etc? It would be preferable if a wider variety of evidence types could be used (also in the table). 6. A further major worry is that some of the case study examples presented have nothing to do with drought or provide no evidence of drought. Yet this paper is specifically dealing with drought. In my view those case studies should not be included. There is considerable mention about severe storms, snow and cold and impacts these have had, but again this is not to do with drought and so only confuses matters further. I strongly suggest that the focus should be much more strongly set on droughts and considerably more evidence presented for such cases. I would have liked to see the inclusion of more quotations that convincingly point to drought conditions. 7. There is no evidence presented for droughts in Canada, yet it features in the table. I suggest that Canada is NOT included in this paper, simply because there is no evidence of Canadian droughts presented in this paper.

I will now work my way through the paper with further general comments and issues to address.

P1, line 29: has the word 'the' too many times. Suggest rather write as: 'This article presents evidence concerning the occurrence and human impacts of. . ..'

P3, line 39: should not say 'in Table 1 below'. . .as there is no table 'below' on that page. . . ..just end it as 'in Table 1.' Same thing on P4 . . .end sentence as '. . .discussion sections.'

Section 3.2.2 Soto Expedition On p5, lines 21/22 you say that the 'only evidence of drought' is based on the fact that people were asked to 'pray for rain to avert a drought and crop failure'. Surely this implies that there was not necessarily a drought yet?....as, by implication, the drought could still be averted. In such a case, maybe the season was drier than normal, hence the request for prayer. . ..but what if the rains came shortly thereafter? So this single line of evidence is not sufficient or convincing for drought conditions. There would need to be other lines of evidence to support this apparent drought.

Section 3.2.3 Coronado Expedition This example presents nothing on droughts at all. In fact, it speaks to above normal winter precipitation. I suggest that this section be cut out.

Section 3.2.4 Luna Expedition Again no droughts here. Famine is reported but seems to be associated with a hurricane in Florida. I suggest this section be cut out.

Section 3.2.5 French and Spanish Florida Colonies Harvest failures are mentioned here but it is not convincingly demonstrated that these are due (only) to drought. Might there have been pests, or poor farming decisions, that contributed to this? Needs further support and some good quotations would help too.

Section 3.2.6 Ajacan Again, there is no convincing evidence from the 'archives of society' that there was drought. Mention is made about 'six years of sterility and

death'…but the text continues to say that the death of plants and crops was due to 'intense cold and snow'. So there is again no strong evidence that drought was the major factor here – it seems more to do with cold and snow. Unless there is stronger supporting evidence for drought, then this section should also be cut out.

Section 3.2.9 Onate expedition Please add some quotations to support the occurrence of drought and elaborate with further examples.

Section 3.2.10 Jamestown Much is also mentioned here about snow and winter cold – not sure of its relevance? We are informed that crops failed repeatedly…but why? Needs a more thorough demonstration as to all factors causing this…or to more convincingly show that it was only due to drought. Maybe poor farming decisions, techniques etc as well? The fact that salt water intruded the James River also does not say much. Was this a normal or abnormal annual (or seasonal) occurrence? Was this due to the river being abnormally low in flow? The context here is missing, or at best vague.

Finally, I am not convinced with the discussion and conclusion which informs us that the archives of society are a good source to classify drought conditions given their general agreement with natural archives. This is simply because the evidence for such droughts is too sparse and lacking in absolute measure. Many of these European Colonial expeditions were on the move and would also not have been able to establish the context of conditions to the longer term 'norm'. The discussion also mentions that the paper addresses the impacts of drought on society, but there is very little in this paper that details precisely this. I suggest a section be written on the impacts of droughts on society during this time period. Overall, a much more convincing case needs to be presented to make this paper work and achieve its aim.

---

## Referee Comment (RC2) · Anonymous Referee #2 · 14 Mar 2019

The paper presents the earliest available documentary evidence for drought for North America, covering the period 1500-1610, a period of early European exploration and colonization. These data are compared to PDSI data from tree rings. It is a valuable and interesting approach presented in a concise manner.

However, there are two major points, that need to be addressed.

The paper is very generous concerning the type of documentary evidence it considers to be an indication of drought. For example in Section 3.2.6 'six years of sterility and

death' are interpreted as evidence for drought, because the tree-ring records show a multi-annual dry period, even though the actual documents only seem to mention 'intense cold and snow' as reasons for the harvest failures. In general the early expeditions (up to about the second half of the sixteenth century) only yield very vague information on the hydroclimate in the explored region, which is not surprising since the members of the expeditions must have been aware of their lack of information to contextualize experienced weather conditions. For the first two expeditions the only evidence for dry conditions are Native Americans asking explorers to pray for rain for averting drought, but it is not clear if this request was merely part of a general seasonal ritual or if it was an indication of dry weather beginning to stress the crops. It seems to be also difficult to precisely date and locate the first example. In Section 3.2.3. 'several mentioned heavy winter snows and none mentioned drought, even though most members of the expedition were expecting a Mediterranean climate' is interpreted as evidence for above-normal winter precipitation. It may be advisable to put more focus on the post-1560 data, because there the evidence for dry conditions is often stronger; this would also offer the possibility to give greater detail for the actual drought descriptions and drought impacts.

The assembled data is actually too sparse to form an outright reconstruction of droughts 1500-1610 – especially considering the vast geographical coverage of the paper – it is more like an assembly of case studies. This is relevant in the comparison to the North American Drought Atlas (NADA) PDSI data. In the discussion the author states: 'In contrast to historians' findings that the corresponding Old World Drought Atlas has been unable to reproduce well-verified historical climate anomalies [...], the NADA appears to consistently identify droughts found in the archives of societies.' The drought information supplied by the case studies presented in this paper is indeed coherent with the NADA PDSI information, but it is in itself not consistent and continuous enough to allow for a systematic comparison. Such a systematic approach should also not only cover the period 1500 to 1610, but the whole pre-instrumental period up to c. 1800 or 1850. This limitation of the data needs to be recognized in more detail.

It should also be added that the representation here of the relationship between hydrometeorological information in European documentary sources or early instrumental observations and the PDSI data in the OWDA is more complex than indicated in this paper. Some extremes are well represented in both types of data, others are not, and the source for the statement in this paper refers only to the decades around 1800, but does not take into consideration the whole available evidence from the Middle Ages onwards, or analyse regional and temporal variation in detail. So the abovementioned phrase should be remodelled to reflect this nuance.

Minor points: Canada. In Table 1 it is clear that the archives of society for Canada have been checked for drought information, but none could be found. This is also indicated in the paper ('evidence for drought and the occurrence of rain prayers in New England and Canada during the 1620s and 1630s (White, 2015a; Grandjean, 2011), which suggests that the absence of evidence during the expeditions under study here likely reflects a lack of observed droughts rather a failure to recognize droughts'). Maybe this lack of drought information in the period 1500-1610 could be somewhat more emphasized – it is easily overlooked – by adding the number of expeditions as well as their names and dates.

Cold and snow: Several times the paper refers to increased cold and snow in winter time, but omits to explain how these conditions would be connected to drought.

p. 1, 23-24. 'for the past five to six centuries in regions with abundant personal records and official archives, such as Western Europe and China': In this time frame a good number of records is available for most parts of Europe.

p. 9, 23-24. 'This suggests that the NADA summer PDSI reconstruction may be more sensitive to summer precipitation at some precise locations': This needs rephrasing.

---

## Author Comment (AC1) · 28 Mar 2019

I'd like to thank the referees for their time in reading the article and for offering their feedback. I have offered a reply to each section of each review below. Because many of the referees' comments overlap, I am posting the full reply to each review.

In general, I am concerned that the reviews misconstrue the goals of the paper, although these were stated in the introduction. This paper is not an attempt to reconstruct drought frequency based solely on historical records. Had it been so, I fully agree

that the available written records would have been inadequate. However, as stated in the introduction, the goals of the paper are: "(1) to test the objectivity and reliability of these historical observations, and thus the potential for exploration and colonization records to be used in drought reconstruction; (2) to crosscheck the NADA reconstructions – including those for extreme events reconstructed during the 16th and early 17th centuries – and the NADA's applicability to the scale of human historical events; (3) to gain further insights into the seasonality and severity of historical droughts found in each type of evidence; and (4) to better understand the human impacts of droughts during this critical and vulnerable phase of North American exploration and colonization." I ask the editors to keep these in mind these stated goals when assessing the applicability of the referees' suggestions and criticisms and the suitability of the paper for publication.

>Reviewer 1

[Referee comments on: Drought during early European exploration and colonization of North America, 1500-1610CE: A comparison of evidence from the archives of societies and the archives of nature This paper takes an interesting and somewhat new and refreshing approach to historical weather/climate studies by comparing so called 'archives of society' (written historical documentary accounts) with those of 'nature' (here mostly tree rings). A great idea I think. The study takes a risk by attempting to establish climatic conditions based on documentary evidence, during what would have been the very beginnings of colo- nial conquest into North America (16th C). This at a time when documentary evidence would have been limited for any given place over any given time (certainly when one compares with the data density for a similar period in a European context). Overall, the paper is stylistically and typologically excellently written and well organized. The abstract suitably covers what it should and is an accurate representation of the paper. I like the introduction (probably the best part of the paper for me) – it is well structured and introduces the topic in a succinct, clear and critical manner. Many of the challenges are highlighted in the introductory sections of

the paper. However, I think some critical challenges are not adequately addressed, and so I will elaborate on some major challenges/concerns that I feel need to be addressed to make this work publishable.]

Although the review has complemented the introduction, I am concerned that the introduction – or perhaps its opening paragraph – somehow misled the referee. The primary goal of the paper was not, as stated in this part of the review "to establish climatic conditions based on documentary evidence." Rather, the primary goal was to test the objectivity and reliability of that documentary record against a well-verified high-resolution reconstruction based on proxies in a natural archive: i.e., the North American Drought Atlas. The paper contributes to climate reconstruction not by using documentary evidence independently to reconstruct drought frequency but instead by helping to establish the validity of evidence drawn from a particular corpus of documents. Once it can be shown that the evidence in this corpus is consistent with our best reconstruction from proxies in natural archives, then historical climatologists and paleoclimatologists may use it, and similar written sources, with greater confidence, whether for independent climate reconstructions or to complement and extend existing reconstructions. Insofar as observations in such documentary records are validated by the comparison, they may supply us information lacking in the natural archives – for instance, the seasonality of droughts and their agricultural and human impacts.

There are several reasons why the records of early colonial North America work well for this purpose: First, for historical records of this era, they are unusually well-preserved and accessible, and they are about as much as one researcher could consult in their entirety (i.e., several tens of thousands of pages, or tens of millions of words). This means that I was able to examine these records contextually and systematically and to ensure that they did not contain significant misleading information pertaining to drought. (And it matters that I could do this personally, because I wouldn't trust research assistants to have done it as thoroughly and consistently across source types.) Surprisingly, not only did historical observers indicate drought in most cases where

they encountered one (in the NADA reconstruction) but also they gave no false positives: that is, they did not ever give strong indications of a drought where one was not present in the NADA record. Second, the NADA is an especially extensive high-resolution reconstruction, providing a good point of comparison for the written historical records. Third, these records represent multiple nationalities, languages, and source types in disparate environments. That fact enabled me to make internal comparisons and to ensure that there was no systematic bias in one source type as compared to another. Fourth, these records are representative of a wider corpus of documentary records of early European exploration and colonization around the world. Such records contain many potentially valuable observations about weather, climates, and environments; yet as described in the paper, they present particular advantages and drawbacks as historical sources. Verifying the reliability of observations about one parameter (drought) in one representative sample (early colonial North America) helps establish the usefulness (or not) of using such documentary records for environmental history and historical climatology more widely.

In the case of early American history, the written information concerning drought turns out to be too sparse to create an independent reconstruction of drought frequency or severity. Nevertheless, the comparison between this evidence and the NADA indicated that the written observations concerning drought presence or absence were remarkably consistent with the tree-ring record. Therefore, it was possible for the paper to address the paper's other three goals as well: to confirm the local agricultural and human impacts of droughts identified in NADA, to indicate the seasonality of those droughts, and to specify how early colonial expeditions were affected. Moreover, by establishing the reliability of information concerning drought, I could more confidently use the other climatic and environmental information in the records, especially the evidence for severe cold during several expeditions.

I would propose to rewrite the first and third paragraphs of the introduction to clarify these points and to remove any expectation that the paper aims to create an independent reconstruction of drought frequency based on this documentary record.

[Major concerns: 1. The paper does not define 'droughts' in the North American context. : : :..and doing so for such a vast region is challenging as 'experiencing dry conditions' in specific areas may not necessarily imply drought, especially if it is during a naturally dry season, or in a dry region. Defining drought is also relative to the individuals' past experience of climate (depending where they came from) – so someone who is accustomed to semi-arid conditions is less likely to identify 'drought' conditions as opposed to someone who is accustomed to a climate of all year plentiful rain. Such context must be taken into account in all instances, which I do not see much of in the case study examples presented in this paper. Then of course there are the different types of drought such as hydrological, climatological, agricultural etc and these differ too, yet the paper is unable to differentiate between these.]

I believe that the paper already addresses each of these points, although I am open to suggestions about how they might be made clearer without the risk of repetition. As explained in the sources section, the NADA is a reconstruction of PDSI. PDSI is a well-known scale among (historical) climatologists and is further described in section 2.2. Section 2.2 also specifies the difference between hydrological, meteorological, agricultural droughts and indicates that evidence in the archives of societies is usually of the latter type. Because the documentary record is not being used to independently reconstruct drought frequency, it would have been misleading to assign a particular standard for "drought" appearing in that record. Since the goal was to test the reliability of that documentary record, it was important to consider all information appearing therein that might indicate the presence or absence of any kind of drought: hydrological, meteorological, or agricultural. Where the documentary record did indicate drought, its type was usually specified in the description (e.g., page 7, line 35). The question of whether foreign observers could accurately assess drought by local standards is explicitly addressed in section 2.1, second paragraph. Indeed, this is one of the central questions which the paper attempts to address by comparing these observations to a more homogenous, continuous reconstruction based on a natural proxy. I propose to make each of these points more explicit in the revised introduction to the paper, and to add qualifiers (hydrological, agricultural, meteorological) to descriptions of drought as appropriate where they may be missing.

[My second major concern is that in the examples presented there is very limited 'societal/historical written evidence' presented to support drought conditions. In most cases there are only one or two lines of evidence and this is surely insufficient, especially given the nature of some of this evidence.]

Again, the goal of the essay is not to independently reconstruct the occurrence of drought from the documentary record but to examine the consistency of all the relevant information in that record with an existing high-resolution reconstruction based on a natural archive. This means that I include information that is not by itself sufficient to prove the occurrence of drought. Had that information proven inconsistent (even if inconclusive) then such inconsistency might have called into question the reliability of the documentary record. What the reviewer has interpreted as overreach in an effort to reconstruct past droughts is really an abundance of caution in an effort to test the documentary record.

In some cases, the paper needs to clarify where the quotations or descriptions provided are only examples of several or many such descriptions and where they are the only such quotations or descriptions in the record. I propose to make this change as necessary.

[This brings me to the third major concern regarding types of evidence used. Famines are of course not necessarily an indication of drought. As the author correctly implies, some of these famines may be due to severe cold, snow, storms, social disruptions etc. But they may also be due to poor farming practices, poor decisions made with regards planting time or most suitable crops, as also pests that might destroy crops. Praying for rain may also not necessarily imply a drought. Prayer might be asked for if the rains

may be delayed, or there may be a mid- summer dry spell etc: : :but if one were to look at the season as a whole it may not have been a drought season. Poor harvests and crop failures may also not imply drought, for some of the same reasons already mentioned above.]

I entirely agree with the referee that by themselves neither famine nor rain prayers should be taken as conclusive evidence of drought. Again, the goal of the essay is not to independently reconstruct the occurrence of drought from the documentary record but to examine the consistency of all the relevant information in that record with an existing high-resolution reconstruction based on a natural record. This means that I include information that is not by itself sufficient to prove the occurrence of a drought. Had that information proven inconsistent (even if inconclusive) then such inconsistency might have called into question the reliability of the documentary record. What the re-viewer has interpreted as overreach in an effort to reconstruct past droughts is really an abundance of caution in an effort to test the documentary record. As it turns out, some famines encountered on these expeditions probably were related drought and some were not. These are each explained in part 3.2. Regarding rain prayers, each observed rain prayer actually did occur in a location and year with PDSI<-1 as recon-structed in NADA, providing a surprising fit between the two types of evidence.. I pro-pose to clarify why each type of information was included in part 2.1, and to emphasize that this information is not by itself conclusive of the presence of drought.

[4. A tough one here, but classifying a season or a year as a drought season or drought year would surely require one to have some sort of bench-mark to compare against (i.e. with other years). An important question for the author to clarify in this regard is whether, based on the documentary evidence, one is able to say whether a season or year is far enough below the 'normal' to define it as a 'drought'. Or, does this paper simply take dry conditions (irrespective of whether it is below normal, normal, or above normal in rainfall/moisture) which affect society, as meaning it is a 'drought'. Better clarity on all this is required.]

The NADA results are on based on the PDSI, thus a standard scale. One point of this exercise is to determine whether these soil moisture deficits reconstructed in NADA were also felt as meteorological and agricultural deficits. The former is admittedly had to judge from the evidence, so I propose to add the qualifier perceived meteorological deficit. The latter is necessarily function of agricultural practices and therefore always relative. In the context of the early colonial American documentary record, agricultural drought can only be determined by reports of success or failure of Native American crops. In this regard, it is also useful to include indications of rain prayers and famines even though, as described above, they are not by themselves conclusive evidence of famine.

[Are there not a wider variety of evidence types that might be discussed in each case study? For instance, reports of grass being dry or sparse, fires, rivers dried up or far below the normal level, death of natural vegetation due directly to drought etc? It would be preferable if a wider variety of evidence types could be used (also in the table).]

Insofar as these types of evidence are phenological, these are included in the phenological evidence. Insofar as they are narrative or descriptive, they are summarized in narrative and descriptive evidence.

[6. A further major worry is that some of the case study examples presented have nothing to do with drought or provide no evidence of drought. Yet this paper is specifically dealing with drought. In my view those case studies should not be included. There is considerable mention about severe storms, snow and cold and impacts these have had, but again this is not to do with drought and so only confuses matters further. I strongly suggest that the focus should be much more strongly set on droughts and considerably more evidence presented for such cases. I would have liked to see the inclusion of more quotations that convincingly point to drought conditions.]

Again, the goal of the essay is not to independently reconstruct the occurrence of drought from the documentary record but to examine the consistency of all relevant

information in that record with an existing high-resolution reconstruction based on a natural archive. In some cases, information regarding snow, storms, etc. is included in order to show that the sources indicated an absence of meteorological drought in a particular season; in other cases, the information is provided to demonstrate that shortages or famines probably occurred for reasons other than drought.

[7. There is no evidence presented for droughts in Canada, yet it features in the table. I suggest that Canada is NOT included in this paper, simply because there is no evidence of Canadian droughts presented in this paper.]

I included Canadian evidence and expeditions to indicate the absence of false positives – i.e., that the documentary record did not falsely report droughts where and when they were absent. When it comes to judging the reliability of the sources, I regard this as equally important to the presence of true positives.

[P1, line 29: has the word 'the' too many times. Suggest rather write as: 'This article presents evidence concerning the occurrence and human impacts of: : :']

Agreed. I will make this change.

[P3, line 39: should not say 'in Table 1 below': : :as there is no table 'below' on that page. : : :.just end it as 'in Table 1.' Same thing on P4 : : :end sentence as ': : :discussion sections.']

Agreed. I will make this change.

[Section 3.2.2 Soto Expedition On p5, lines 21/22 you say that the 'only evidence of drought' is based on the fact that people were asked to 'pray for rain to avert a drought and crop failure'. Surely this implies that there was not necessarily a drought yet?....as, by implication, the drought could still be averted. In such a case, maybe the season was drier than normal, hence the request for prayer: : :.but what if the rains came shortly thereafter? So this single line of evidence is not sufficient or convincing for drought conditions. There would need to be other lines of evidence to support this

apparent drought.]

Again, I regarded it as important to disclose all relevant information in order to ensure there were no false positives. In this case, there is not clear evidence whether the Indian nation in question was already suffering a crop failure or famine. However, the tree ring record indicates that there was a (PDSI) drought that year and that it was the only region encountered by the Soto expedition with such a drought. Is it not worth noting that this is the only indication of drought during the entire Soto expedition? If the sources were unreliable or if rain prayers were not indicative of drought, wouldn't that be an extraordinary coincidence? I could emphasize this point more in the text.

[Section 3.2.3 Coronado Expedition This example presents nothing on droughts at all. In fact, it speaks to above normal winter precipitation. I suggest that this section be cut out.]

Again, the point was to demonstrate the absence of false positives. Compare the Coronado and Soto expeditions: If the (New) Spanish were simply inclined to describe the Southwest as dry (because it is drier than Spain) and were incapable of distinguishing relatively wet and dry years for that region's climate, then we would expect similar descriptions from both expeditions. Instead, the Coronado expedition leaves no evidence of drought while the Oñate expedition leaves abundant evidence of drought. I could emphasize this point more in the text.

[Section 3.2.4 Luna Expedition Again no droughts here. Famine is reported but seems to be associated with a hurricane in Florida. I suggest this section be cut out.]

See comment for Coronado expedition.

[Section 3.2.5 French and Spanish Florida Colonies Harvest failures are mentioned here but it is not convincingly demonstrated that these are due (only) to drought. Might there have been pests, or poor farming decisions, that contributed to this? Needs further support and some good quotations would help too.]

It's unclear whether the French Florida famines were due to drought. Famines in parts of early Spanish Florida were specifically blamed on drought, and so I will emphasize that in the text.

[Section 3.2.6 Ajacan Again, there is no convincing evidence from the 'archives of society' that there was drought. Mention is made about 'six years of sterility and death': : :but the text continues to say that the death of plants and crops was due to 'intense cold and snow'. So there is again no strong evidence that drought was the major factor here – it seems more to do with cold and snow. Unless there is stronger supporting evidence for drought, then this section should also be cut out.]

Again, the point is not to prove from documentary sources that there was a drought. The soil moisture deficit is already strongly indicated in the tree ring record. The point is simply to demonstrate the compatibility of the two types of evidence and to show that there is not a false positive in the written record.

[Section 3.2.9 Onate expedition Please add some quotations to support the occurrence of drought and elaborate with further examples.]

In this case, there are ample descriptions of the drought during 1599-1600. I will provide a specific quotation or two and indicate that these are only one or two of many examples.

[Section 3.2.10 Jamestown Much is also mentioned here about snow and winter cold – not sure of its relevance? We are informed that crops failed repeatedly: : :but why? Needs a more thorough demonstration as to all factors causing this: : :or to more convincingly show that it was only due to drought. Maybe poor farming decisions, techniques etc as well? The fact that salt water intruded the James River also does not say much. Was this a normal or abnormal annual (or seasonal) occurrence? Was this due to the river being abnormally low in flow? The context here is missing, or at best vague.]

I regarded it as important to mention the extreme cold in order to indicate that there might have been other reasons for crop failures and famine. Settler crop failures were probably due to poor decisions, but it is unlikely that Native American crop failures were due to similar poor decisions. The intrusion of saltwater up the James River is an unusual occurrence, and I will clarify that in the text.

[Finally, I am not convinced with the discussion and conclusion which informs us that the archives of society are a good source to classify drought conditions given their general agreement with natural archives. This is simply because the evidence for such droughts is too sparse and lacking in absolute measure. Many of these European Colonial expeditions were on the move and would also not have been able to establish the context of conditions to the longer term 'norm'. The discussion also mentions that the paper addresses the impacts of drought on society, but there is very little in this paper that details precisely this. I suggest a section be written on the impacts of droughts on society during this time period. Overall, a much more convincing case needs to be presented to make this paper work and achieve its aim.]

Again, the reviewer here assumes that the aim of the paper is to reconstruct drought frequency using the archives of societies and, therefore, that it falls short of its goal. However, as emphasized in this response, that is not the goal of the paper. As specified in the introduction, the primary aim of the paper is to test the accuracy of the historical records against a high-resolution tree ring-based drought reconstruction. Therefore, the referee's concern is not applicable. In fact, the referee's statement indicates why this paper is important after all. As stated previously, the documentary record of early European exploration and colonization – not only in North America but around the world – contains a wealth of observations about environments and climates. At the same time, these observations were made under atypical circumstances by often inexperienced observers. For those reasons, one might be inclined to dismiss the information out of hand. A more productive approach, I believe, is to find some parameter for which we can systematically test the observations in a documentary record against an

independent reconstruction from a natural proxy. Drought in early colonial North America happens to provide such a test. It could well have turned out that the documentary record frequently contained indications of drought where and when there wasn't any, or that observers failed to notice most of the droughts they did encounter. In fact, it turns out the information in the archives of societies is quite compatible with the reconstruction from natural archives. That doesn't mean we have to accept everything colonial observers tell us. However, it does indicate that we should take their descriptions of weather and environmental conditions seriously.

>Reviewer 2

[The paper is very generous concerning the type of documentary evidence it considers to be an indication of drought. For example in Section 3.2.6 'six years of sterility and death' are interpreted as evidence for drought, because the tree-ring records show a multi-annual dry period, even though the actual documents only seem to mention 'intense cold and snow' as reasons for the harvest failures. In general the early expeditions (up to about the second half of the sixteenth century) only yield very vague information on the hydroclimate in the explored region, which is not surprising since the members of the expeditions must have been aware of their lack of information to contextualize experienced weather conditions. For the first two expeditions the only evidence for dry conditions are Native Americans asking explorers to pray for rain for averting drought, but it is not clear if this request was merely part of a general seasonal ritual or if it was an indication of dry weather beginning to stress the crops. It seems to be also difficult to precisely date and locate the first example. In Section 3.2.3. 'several mentioned heavy winter snows and none mentioned drought, even though most members of the expedition were expecting a Mediterranean climate' is interpreted as evidence for above-normal winter precipitation. It may be advisable to put more focus on the post-1560 data, because there the evidence for dry conditions is often stronger; this would also offer the possibility to give greater detail for the actual drought descriptions and drought impacts.]

The paper does not describe these observations as an "indication of drought" but only as "evidence concerning drought". As with reviewer 1, there is an underlying assumption here that the paper is aiming at a reconstruction of drought frequency based on historical records. It is not. On the contrary, it is primarily a test of those records utilizing a comparison to a high-resolution tree ring-based drought reconstruction. I will make this point clearer in the introduction, as indicated above.

As discussed in the response to reviewer 1, the goal of the essay is not to independently reconstruct the occurrence of drought from the documentary record but to examine the consistency of all the relevant information in that record with an existing high-resolution reconstruction based on a natural archive. This means that I include information that is not by itself sufficient to prove the occurrence of a drought. Had that information proven inconsistent (even if inconclusive) then such inconsistency might have called into question the reliability of the documentary record. What the reviewer has interpreted as overreach in an effort to reconstruct past droughts is really an abundance of caution in an effort to test the documentary record.

[The assembled data is actually too sparse to form an outright reconstruction of droughts 1500-1610 – especially considering the vast geographical coverage of the paper – it is more like an assembly of case studies. This is relevant in the comparison to the North American Drought Atlas (NADA) PDSI data. In the discussion the author states: 'In contrast to historians' findings that the corresponding Old World Drought Atlas has been unable to reproduce well-verified historical climate anomalies [...], the NADA appears to consistently identify droughts found in the archives of societies.' The drought information supplied by the case studies presented in this paper is indeed coherent with the NADA PDSI information, but it is in itself not consistent and continuous enough to allow for a systematic comparison. Such a systematic approach should also not only cover the period 1500 to 1610, but the whole pre-instrumental period up to c. 1800 or 1850. This limitation of the data needs to be recognized in more detail.]

At no point does the paper claim that the documentary data can form an outright reconstruction of drought frequency or severity. The paper provides a systematic review of the nearly the entire documentary record for the United States and Canada during this period. If that documentary record were so comprehensive as to include possible observations for every drought in every location during an entire century, then it would have been far too vast to investigate thoroughly and consistently. That thoroughness is key, because it enables the paper to demonstrate the consistency of the entire corpus of evidence with the tree-ring record, hence demonstrating (so far as possible) the reliability (if not completeness) of the meteorological and environmental observations in these records. Again, since this is not a climate reconstruction, but primarily a test of the written evidence, what matters is not a systematic coverage of every drought but a systematic coverage of all the written evidence against the most reliable and high-resolution reconstruction from the archives of nature.

[It should also be added that the representation here of the relationship between hydrometeorological information in European documentary sources or early instrumental observations and the PDSI data in the OWDA is more complex than indicated in this paper. Some extremes are well represented in both types of data, others are not, and the source for the statement in this paper refers only to the decades around 1800, but does not take into consideration the whole available evidence from the Middle Ages onwards, or analyse regional and temporal variation in detail. So the abovementioned phrase should be remodelled to reflect this nuance.]

I agree and will change this sentence accordingly.

[Minor points: Canada. In Table 1 it is clear that the archives of society for Canada have been checked for drought information, but none could be found. This is also indicated in the paper ('evidence for drought and the occurrence of rain prayers in New England and Canada during the 1620s and 1630s (White, 2015a; Grandjean, 2011), which suggests that the absence of evidence during the expeditions under study here likely reflects a lack of observed droughts rather a failure to recognize droughts'). Maybe this lack of drought information in the period 1500-1610 could be somewhat more emphasized – it

is easily overlooked – by adding the number of expeditions as well as their names and dates.]

The names, dates, and locations of those expeditions are already included in Table 1. Since Reviewer 1 has left the opposite advice concerning the mention of these expeditions to Canada and New England, I will defer to SI editor's judgement on this matter, or otherwise leave it as is.

[Cold and snow: Several times the paper refers to increased cold and snow in winter time, but omits to explain how these conditions would be connected to drought.]

The discussion of cold and snow would seem irrelevant only if the reviewer assumes that this paper is a reconstruction of drought frequency and severity based on the written records, rather than a test of those records. The point in mentioning them here is to include all relevant information concerning the presence, absence, impacts, or seasonality of drought. As discussed in the paper, the NADA reconstructs soil moisture, which is a function of precipitation and evapotranspiration in all seasons. Cold, snowy winters are thus significant for at least three reasons: (1) Assuming the historical evidence is accurate, a cold snowy winter indicates that any drought indicated for that year and location in NADA should have arisen from a summer precipitation deficit. (2) If the historical sources mention a cold snowy winter but don't provide any indication of summer rainfall deficit, and yet the NADA reveals a drought for that year, that would tend to suggest that the weather observations in the historical sources are inaccurate. (3) Under certain circumstances, cold snowy winters provide a potential explanation for crop failure or famine other than drought.

[pp. 1, 23-24. 'for the past five to six centuries in regions with abundant personal records and official archives, such as Western Europe and China': In this time frame a good number of records is available for most parts of Europe.]

I agree and will change the sentence accordingly.

[p. 9, 23-24. 'This suggests that the NADA summer PDSI reconstruction may be more sensitive to summer precipitation at some precise locations': This needs rephrasing.]

I agree and will change the sentence.

---

## Author Response (AR1)

**Editor Decision: Reconsider after major revisions** (19 Apr 2019) by Andrea Kiss
Comments to the Author:
The paper is based on an impressive amount of research work, and provides important, systematically collected, early information regarding droughts or dry conditions reported in documentary evidence, from the earliest period of European colonisation in different parts of North America. The reviews also emphasise the importance of this work, appreciate the ideas, and present an overall positive opinion about the paper. However, there are some very useful major and minor suggestions in the reviews that the author should consider in the revised version of his manuscript. Additionally, I also have some further comments, suggestions to the paper – partly independent, but mostly related to the reviewers' comments and suggestions.

I thank the reviewers and editor for their thoughtful comments and suggestions. Based on the reviewer and editor comments and suggestions, it seemed appropriate to completely revise the paper. I have reconsidered the classification and evaluation of evidence for several expeditions, reviewed certain records for relevant information, entirely revised the summary of drought information into two new tables, and substantially re-written parts of the paper, including: a new section 1, substantial revisions to sections 2.1 and 2.2, a restructured and revised part 3, a new section 4, and new section 5 (incorporating material formerly in section 4).

Independent of the reviewers' comments, I have some doubts regarding the very general use of "archives of societies" and "archives of nature" in the title. Based on the content of the paper, under "archives of societies" you exclusively mean European colonial documentary evidence, and this is compared in the paper to the "archives of nature", and under the later term you predominantly mean tree-ring based SPDI evidence and, in two cases, you also refer to stalagmite evidence. Based on the title, it is not very clear what evidence you will include in the paper. In fact, the term "archives of societies" in comparison with "archives of nature" would suggest that you deal with all source evidence primarily related to human activities (i.e. documentary and archaeological evidence) on the one hand, compared to all the available natural scientific evidence regarding this period on the other. However, "archives of nature" in this sense should include, for example, sedimentary evidence, too: the paper examines a period of over hundred years, and this period is long enough to apply the available archaeological and, especially, sedimentary evidence, but no such evidence is applied in the paper. To overcome this problem, I would suggest to modify the title so that it more adequately reflects the content (e.g. "a comparison of documentary and high-resolution natural-scientific evidence" or "with tree-ring and stalagmite evidence"; essentially, with including the term "high-resolution", one does not miss so much any more the testimony of archaeological and sedimentary evidence, although in both cases sometimes it is possible to find high-resolution information).

I have changed the title to: "A Comparison of Drought Information in Early Colonial Documentary Records and a High-Resolution Tree Ring-Based Reconstruction". I have eliminated any direct comparison between the information documentary record and non-NADA proxy-based reconstructions, except briefly in the discussion.

Another comment reflects on the use of source quality references throughout the paper. For example, although you do not mention (and would be very beneficial to do so), it indirectly becomes clear that you predominantly applied contemporary sources, descriptions of eye-witnesses. It would more clearly demonstrate the quality of the applied source evidence if

you reflect on this question in one or two sentences (and some actual examples), in Sect. 2.1, and also in each case study.

I have added further discussion of the quality of the relevant sources to each part of the Results section, and I have rewritten the first two paragraphs of the introduction to address this point. I have eliminated discussion of observations that were not first-hand, except to note that these can support first-hand descriptions.

Moreover, as suggested in the Introduction, you also applied in two cases stalagmite evidence – its brief presentation, however, is missing from Sect. 2.2.

Given that tree ring-based reconstructions may have problems capturing low-frequency variability (the so-called "segment length curse") I have looked up speleothem or other low-resolution data where it may be useful to confirm (or potentially refute) reconstructed hydroclimate anomalies on a decadal scale. I have added a comment to that effect in section 2.2.

Regarding the Results (Chapter 3), a further comment is related to the case studies: whereas presentation of tree-ring and speleothem evidence always accompanied by very adequate source/literature references, the presentation of documentary evidence usually lacks any direct reference within the text on the sources, but sometimes one-one reference, occasionally, is included. Reference practice should be consequent within the case studies: i.e. please, include the relevant references (i.e. the source reference on drought) also in the description of documentary findings. In relation to the applied source evidence, you provided an Appendix of impressive length on source references. However, now the source references are in a simply alphabetic order, but – as you did not provide the related references within the documentary-based results - it is now difficult to use, and practically impossible to connect to the case studies by simple reading. I would suggest to group the full literature references of the applied sources, provided in the Appendix, to group according to the expeditions, described in Chapter 3. In this way the reader can easily connect the references, now added in Chapter 3, to the actual entries, listed in the Appendix.

Since many of the publications contain multiple sources discussing multiple ventures, it would be difficult to classify them in the way proposed. I have instead numbered the references in the appendix and included a column in the table with the numbers of the most important references.

Furthermore, in the case studies sometimes such terms are used as "second-hand" or "unreliable" source: this is not enough – in these cases, please, clearly define what you mean, and refer to the source (is it a non-contemporary source? Or contemporary, but only heard the information from eye-witnesses or others?). As your sources predominantly originate from contemporary eye-witnesses (i.e. high source quality evidence), it is also your interest to define those exceptional cases when you consider less reliable evidence.

I have added further discussion of the quality of the relevant sources to each part of the Results section, and I have rewritten the first two paragraphs of the introduction to address this point. I have eliminated discussion of observations that were not first-hand, except to note that these can support first-hand descriptions.

Another comment concerns the affected regions, the approximate itinerary and locations

mentioned in the case studies, based on documentary evidence. While a notable amount of figures and maps are included in the paper from the NADA database there is no map that shows the locations/regions mentioned in the documentary case studies. Please, provide either at least an overview map of the locations, regions and itineraries or include this evidence on the related NADA maps (where it would be anyway beneficial to have some orientation points, it is especially true for the New Mexico NADA maps).

I have added approximate routes and the locations of principal colonies to the maps, when sufficient information was available.

A final comment concerns the Discussion and Conclusion: here it seems that the Discussion contains the conclusions, while in the Conclusion some of the conclusions, already mentioned in the Discussion, are repeated. In the Discussion there is a possibility to discuss certain significant questions, possible contradictions, large-scale comparisons etc. in more details.

I have moved the material formerly in the Discussion section to the Conclusion and re-written both sections. The Discussion now addresses patterns in, and questions raised by, the Results.

For example, you raise a rather interesting problem with mentioning the 1770-1772 crisis in Central Europe, and the conclusions on the OWDA-documentary comparison – this question, with using other previous comparisons of OWDA (e.g. Pfister et al., Bauch, Dobrovolny et al. Kiss etc.) – or especially if there is any other comparison with NADA – and documentary evidence, would be a rather interesting and useful topic for further discussion. And, what you now included in the Discussion chapter, could mainly go into the Conclusion.

I have substantially expanded the discussion the comparison between the NADA and OWDA, noting difference in their coverage and reconstruction skill. I have added a reference to a comparison of documentary and OWDA reconstructions (Kiss, 2017). Throughout the paper I now address the difference in reconstruction skill for different regions in the NADA.

Some reflections related to the comments of the Reviewers and the Author's responses: Reviewer No. 1 addresses many important points that can help improving the manuscript. Dear author, please, take into consideration the suggestions while reworking the paper. Moreover, some of the problems, addressed in the general comments of the reviewer could be rather interesting, further topics for the Discussion chapter. Regarding the highly relevant question of drought definitions, it would be useful to provide a short description of what is considered in the discussed regions as drought, and – if there is any information on that – what drought meant in the referred historical documentation, in the Methodology chapter.
This means the inclusion of, for example, the modern understanding what drought is in the relevant areas (i.e. case studies), including the different drought types, and also its historical understanding. Here you could also add what type of dryness or drought-related information types are known from the sources you used in the present work. The interpretation of drought-related documentary information types is of crucial importance to the understanding of the case studies in Sections 3.2.

In the introduction, I now explicitly acknowledge, whereas the tree ring-based reconstruction identifies soil moisture deficits, the documentary records identifies seasonal meteorological, hydrological, or agriculture droughts. I have tried to be clearer about what was observed in each expedition described in section 3.2, adding further quotations and descriptions where relevant.

While the NADA database provides continuous drought-related reconstruction, the available documentary evidence is a fragmentary set of information, an "assembly of case studies". Therefore, based on this comparison, no general conclusion can be drawn on whether or not in all cases the tree-ring evidence fit the documentary evidence. Based on your answers to the reviewer, I would like to ask, please, take the time, and read after what exactly the NADA reconstruction is based on, what information it provides, and what not. In your paper, and especially in your reflections on the reviews you refer to tree-ring based reconstructions like the NADA and the OWDA as they would provide the ultimate truth about droughts, their timing and intensity, and if the documentary evidence does not "fit" them, then they provide false information. This view also reflects in the conclusion when you are criticizing another work for suggesting a contradiction between documentary evidence and the OWDA reconstruction.

In the introduction, I have added a discussion of the value of comparing the documentary information and the reconstruction, making clear that one is not an objectively "correct" reconstruction, and clarifying what can and cannot be learned from this comparison.

Tree-ring evidence is already in itself dependent on a number of environmental factors (i.e. not only precipitation, and not only of specific seasons), and even if it shows relatively high correlation with precipitation (those which shows) this relationship is usually not consistent – changing in time. Tree ring evidence, when primarily related to precipitation, is not available everywhere: for large regions no any evidence is available at al. To overcome this problem, the NADA, OWDA reconstructions contain extrapolated information, which can provide information even in large areas without any tree-ring evidence; nevertheless, the accuracy of the reconstruction (e.g. how accurately they capture the intensity of drought) might be lower than in the areas with high density of information. Concerning Europe, already a number of studies (not only Collet) suggested clear differences between documentary-based drought reconstruction and the OWDA reconstruction. In most cases these studies discussed periods and areas with very extensive, detailed and case-sensitive documentation – with a coverage much denser than the availability of tree-ring evidence. Thus, on the example of Europe, if the tree-ring based reconstruction shows a somewhat different picture than a documentary-based reconstruction, this does not automatically mean that the documentary evidence is "wrong". Not talking about the fact that documentary evidence often provides local information, while tree-ring based overviews mainly concentrate on large-scale general information. Naturally, the situation might be different in North America; however, if you think this is the case, you have demonstrate it first. I kindly suggest to look after how drought information is derived from tree-ring evidence, and also how e.g. the NADA reconstruction was made (you can also find there references on individual tree-ring reconstructions, so you can app. even know about the density of available tree-ring evidence in the areas you have discussed), and what "well-verified" information means in that context.

In the introduction, I have added a discussion of the value of comparing the documentary information and the reconstruction, making clear that one is not an objectively "correct" reconstruction, and clarifying what can and cannot be learned from this comparison. In the Discussion, I have further addressed possible reasons why the NADA and OWDA may achieve different results when compared to local historical observations. For instance, the OWDA uses only 106 tree-ring chronologies compared to 1,845 tree-ring chronologies in the NADA. I have eliminated the term "well-verified" and explained in the Materials section that past studies have also compared NADA reconstructions with historical observations and found that they were consistent.

Moreover, it is worth to check some tree-ring based hydroclimate reconstruction papers regarding the studied regions, because these papers also individually refer to the particular season(s) on which the reconstruction primarily reflects. Thus, in this way you can avoid generalisations, and can actually tell in which of your studied regions the tree-rings react more on winter and in which region e.g. on summer precipitation.

I have checked for specific regional information on the seasonality of tree-ring signals and noted this in section 2.2.

Detailed individual comments of the reviewer with page numbers:
I think the reviewer's suggestions are highly relevant concerning pages 1 and 3, and I find particularly useful and important all the comments related to the Results chapter, particularly the ones reflecting on the case studies.

(These suggestions are reflected in the changes already noted above.)

Regarding the Sections the reviewer suggested to remove: I fully agree that 3.2.3, 3.2.4 should be removed as in the documentary evidence there is no any indication of a drought, and so no comparison can be presented. However, I think you could perhaps utilise some of the information from 3.2.3 in 3.2.2 (as a comparison or additional information?); similarly, some of the information in 3.2.4 could be utilised perhaps in 3.2.5. I agree with the comments regarding 3.2.6: please, provide more details on droughts or if there is no more evidence, why and what extent you connect this "sterility and death" to drought. Please, do consider the suggestions of the reviewer concerning Sections 3.2.7.10.

I have entirely restructured section 3 in order to address these concerns.

In general, many of the problems described by the reviewer could be avoided if the author provides more information on the source, not only with referring, but also on how many sources reflect on drought. My general impression is that in most cases probably only one or two short references are available, while in a few cases (esp. towards the end of the study period) "abundant" information is available. Please, give more exact information. What does "abundant" mean? What does each source suggest? If they are too many to describe one by one, you can group them. It would be generally good to get exactly the information included in the sources. Same goes for dating: we only get information on the year when the report comes from – is it possible that no more information is provided in these sources? Because, of course, the season of report (date) would be often highly relevant. Please, when it is possible, do give the exact date of report, and not only the year.

I have indicated in the table whether information came from a single or multiple sources. It is not possible to give an exact date of each report because the source types are heterogeneous, but I have noted in the text cases when information was only recorded years after the events described.

I agree with the reviewer that quotation would also help in the more accurate presentation of the cases. It would also help the reader if you provide a short conclusion on what exactly the sources tell us about the drought, and how certain, based on the documentary evidence, is that really a drought occurred. You provide excellent conclusion in each case concerning the results derived from tree-ring evidence. It would be much easier to compare the two different set of data if interpretation and conclusion would be available on both sides and not only on the tree-ring (or speleothem) side.

I have entirely revised table 1 and created a new table (table 2) summarizing the results of the comparison by year and location.

Concerning the final conclusions, the reviewer raises a rather interesting question – worth for, for example, another topic to elaborate in the Discussion chapter. I agree with the reviewer that in the Discussion/Conclusion the conclusions are too general and too "overall" compared to the fact that here we are talking about less than a dozen case studies, sometimes with rather sporadic documentary evidence. It does not necessarily means that the conclusions are wrong, but they should be phrased differently, more concentrating on the fact that we are not talking about the conclusions of a 5-6 hundred-year systematic comparison, but only some cases from the 16th and the early 17th centuries.

I appreciate this concern and accordingly have made the Discussion more cautious and specific.

I find the suggestions of the reviewer regarding a separate section on societal impacts highly relevant: if one of your main goals are to present societal impacts, then this topic must be much more emphasised and systematically discussed within the Results chapter. Additionally, you also found it an important aim to "gain further insights into the seasonality and severity of droughts" – it is true, there are some short reflections on these questions, but especially on seasonality you could provide more information. If society, seasonality and severity detection are major goals, then they should systematically appear (or the statement that no information is available)) in the discussion conclusions of each case studies. Moreover, the major results related to your major goals should as well appear in the final conclusion.

I have expanded the discussion of societal impacts. However, it is not one of the main goals of this paper to discuss drought impacts:
1) This is not an article about drought per se, but rather about using drought records to examine the reliability and applicability of certain a certain documentary record and a tree ring-based drought reconstruction.
2) Insofar as the documentary record appears to be reliable, it suggests that the greatest societal impacts arose from unexpected severe cold and storms, with drought often playing a secondary role.
3) I already wrote an entire book about the role of climate in European exploration and colonization of North America. A section on impacts here would neither adequately summarize that book nor add meaningfully to its findings.

Comments regarding Reviewer No. 2.

Most of the major points, mentioned by the second reviewer, are already addressed in connection with the first review, so I only reflect on questions that differ from the comments of Reviewer No. 1, and/or the second reviewer discussed it in more detail than the first.

Minor points: I agree with both reviewers that, since no any drought was described in documentary evidence in Canada during this period, Canada should be taken out of the referred regions. However, I do not see any problem to keep North America in the title, and I think there is no reason to take Canada off from Table 1. Additionally, in the Introduction I find it important to keep the information that the author indeed studied the sources referring to Canada, but no drought-related evidence was found in contemporary documentary evidence (i.e. "lack of observed drought") – with the additions suggested by the second reviewer.

I have removed Canada and New England from table 1 and restructured section 3 to address this issue.

[revised manuscript text omitted]

---

## Author Response (AR2)

[revised manuscript text omitted]

drought.memphis.edu

**Responses to Reviewer**

I appreciate that the title is already quite long but as it stands one might expect to read a paper addressing a global overview when in fact the paper is for North America. The spatial context (i.e. N America) should thus be reflected in the title. One possibility might be: '……in Early North American Colonial Documentary……'

*Accepted and revised.*

Should it be 'a High-resolution Tree Ring-Based Reconstruction' or 'High-resolution Tree Ring-Based reconstructions' ?

*The only systematic comparison is made with the NADA, and therefore, I will maintain the singular.*

2. P3, lines 39/40: avoid using the concept 'archives of societies'

*Accepted and revised.*

3. P6, line 10:….should 'but no left' read as 'but not left' ?

*Accepted and revised.*

4. P6, line 33: '….indicates slightly dry…..'

*Accepted and revised.*

5. P6, line 44: was it a specific shipwreck? If so, then need to mention the ship by name or at least say 'a shipwreck'…otherwise it should read as 'shipwrecks'

*Accepted and revised.*

6. P9, line 1: does it really say "at flood very salt….' ? I ask because this is strange English.

*Yes, this is the original wording.*

7. P10, line 25: avoid the concept 'archives of societies'

*Accepted and revised.*

8. P10, line 26: should read as '…droughts in North America during…..'

*Accepted and revised.*

9. P10, lines 33-36: this is one very long sentence which does not quite connect where you have the words 'and thus to inform'. At that point it may be better to start a new sentence: 'Hence, this informs……'

*Accepted and revised.*

10. P11, lines 8 & 9: you have the word 'certain' four times in this sentence – this is not appropriate ….look for appropriate synonyms

*Accepted and revised.*

11. P11, line 18: should be 'findings'

*Accepted and revised.*

**List of changes**

Title:
A Comparison of Drought Information in Early **North American** Colonial Documentary Records and a High-Resolution Tree Ring-Based Reconstruction

p3, line 39: "compared to other similar written records"

p4, line 12: "conditions of rivers, provide"

p6, line 10: "but left no definite indications"

p6, line 33: "1559-1560 (Figure 2a) indicates slightly dry conditions"

p6, line 40: "over-run in 1565 by Spanish soldiers, who"

p6, line 44: "poor supplies, shipwrecks, and inability"

p9, line 6: "did not swim up the James River as usual, which"

p9, line 16: "It indicates the years and regions where"

p10, line 25: "This study finds broad agreement between the evidence in written records and the tree-ring based NADA concerning"

p10, lines 34-35: "pathogens (e.g., White, 2014; Wickman, 2018). Hence, this informs studies of proto-historic climate impacts"

p11, line 8: "has a high probability"

p11, line 9: "and a low probability"

p11, line 11: "presence or absence of those observations in the records for a given time and place might be used to derive posterior"

p11, line 18: "previous research findings that drought"

¶